# Self-Assembling Tacrolimus Nanomicelles for Retinal Drug Delivery

**DOI:** 10.3390/pharmaceutics12111072

**Published:** 2020-11-10

**Authors:** Vrinda Gote, Abhirup Mandal, Meshal Alshamrani, Dhananjay Pal

**Affiliations:** Division of Pharmaceutical Sciences, School of Pharmacy, University of Missouri-Kansas City, 2464 Charlotte Street, Kansas City, MO 64108, USA; vrindagote@mail.umkc.edu (V.G.); abhirupmandal@mail.umkc.edu (A.M.); malshamrani@jazanu.edu.sa (M.A.)

**Keywords:** retinal pigment epithelial cells, sodium iodate, pro-inflammatory cytokines, reactive oxygen species, VEGF-A

## Abstract

Neovascular age-related macular degeneration (AMD) is characterized by an increase in reactive oxygen species (ROS) and pro-inflammatory cytokines in the retinal pigment epithelium cells. The primary purpose of this study was the development of a clear, tacrolimus nanomicellar formulation (TAC-NMF) for AMD. The optimized formulation had a mean diameter of 15.41 nm, a zeta potential of 0.5 mV, and an entrapment efficiency of 97.13%. In-vitro cytotoxicity studies revealed the dose-dependent cytotoxicity of TAC-NMF on various ocular cell lines, such as human retinal pigment epithelium (D407), monkey retinal choroidal endothelial (RF/6A) cells, and human corneal epithelium (CCL 20.2) cells. Cellular uptake and in-vitro distribution studies using flow cytometry and confocal microscopy, respectively, indicated an elevated uptake of TAC-NMF in a time-dependent manner. Biocompatibility assay using macrophage RAW 264.7 cell line resulted in low production of inflammatory cytokines such as IL-6, IL-1β and TNF-α after treatment with TAC-NMF. There was a decrease in ROS in D407 cells pre-treated with sodium iodate (ROS inducing agent) after treating with TAC-NMF and tacrolimus drug. Similarly, there was a reduction in the pro-inflammatory cytokines and VEGF-A in D407 cells pretreated with sodium iodate. This indicates that TAC-NMF could lower pro-inflammatory cytokines and ROS commonly seen in AMD.

## 1. Introduction

Age-related macular degeneration (AMD) is one of the most common back of the eye disorders in the United States [1,2,3,4]. AMD can cause severe vision impairment and lead to vision loss. It is one of the most imperative causes of blindness in people above 55 years of age, and it is estimated that more than 3.5 million people will be affected by this ocular disorder in the United States by 2030 [5]. The current treatment for wet AMD includes use of anti-VEGF monoclonal antibodies such as ranibizumab and aflibercept [6]. Although highly effective, each intravitreal injection has the risk of adverse ocular events like endophthalmitis, retinal detachment and retinal hemorrhage [7,8].

Retinal pigment epithelium (RPE) is a specialized metabolically active layer of the retina which provides energy to the photoreceptor cells, helps in its protection of photo oxidation and helps in the phagocytosis of the dead photoreceptors. AMD is characterized by an increase in reactive oxygen species (ROS) and pro-inflammatory cytokines in the RPE layer. This results in a cascade of inflammatory response and development of tissue atrophy [9,10]. Ocular tissue and ocular fluids of patients with AMD have shown elevated levels of IL-6, IL-8, and TNF-α [11]. Similarly, the in-vitro upregulation of IL-6, IL-8 and CCL2 from ARPE-19 cells was observed after treatment with cytotoxic agents [12]. Various pro-inflammatory cytokines and chemokines activate the nuclear factor kappa-B (NF-kB) transcription factor. This further aggravates inflammation [13]. Inflammation in the RPE can lead to infiltration of macrophages and T cells [14]. This further augments to upregulation of COX-2, ICAM-1, caspase 1, iNOS, nitric oxide, IL-1β, prostaglandin E2, VEGF-A, NF-kB and cytokines production [9,15]. VEGF-A upregulation leads to production of new blood vessels leading which are fenestrated and leaky [16]. Elevated oxidative stress followed by inflammation and finally angiogenesis results in AMD progression [17].

Tacrolimus (FK 506) is a potent immunosuppressive macrolide drug. It used for atopic dermatitis and rheumatoid arthritis [18,19,20]. Tacrolimus is a hydrophobic drug with a Log P value of 2.7 and molecular weight of 804 g/mol [21]. The drug is also being currently investigated for treatment of ocular diseases like dry eye disease, vernal keratoconjunctivitis and ocular graft-versus-host disease in various clinical trials (NCT01850979, NCT00567762, NCT01977781). Tacrolimus has also shown to down-regulate various inflammatory markers and NF-kB pathway in various cell lines [22,23,24,25]. The drug also has been shown to prevent early retinal neovascularization in mice ocular tissues which were treated with streptozotocin-induced diabetic retinopathy [26].

Ocular drug delivery using nanomicelles has been widely explored for various ocular indications due to the advantages like controlled drug release, effective crossing of ocular barriers and improved bioavailability with minimal ocular toxicity [27,28]. Nanomicelles, due to their unique structure, can encapsulate highly hydrophobic drugs and enhance their solubility. This can facilitate in drug penetration and effective drug delivery to the target tissue. Mixed amphiphilic nanomicelles demonstrate a stronger hydrophobic interactions and stronger hydrogen bonding with the adjacent polymers [29]. In this study, we have utilized a mixture of two polymers; PEG-Hydrogenated Castor oil-40 (HCO-40) and octyxonyl-40 (OC-40) for the nanomicellar formulation. These polymers are classified as “Generally Regarded as Safe” (GRAS) by the US-FDA. Amphiphilic nanomicelles can form a clear aqueous solution of hydrophobic drugs, suitable for ophthalmic solutions [30,31]. Clear nanomicellar solutions of hydrophobic drugs can be suitable for topical administration and for intravitreal administration due to their clarity [28,32].

The present study aimed at developing a clear and stable nanomicellar formulation of tacrolimus to reduce inflammation and oxidative stress in RPE, which is the first sign of macular degeneration. For this, we developed a nanomicellar formulation of tacrolimus using an optimized mixture of two polymers, HCO-40 and OC-40. Tacrolimus nanomicellar formulation (TAC-NMF) was optimized by using full factorial design of experiment in JMP^®®^ Design of Experiment (DOE) software. Nanomicellar size, poly dispersity index, zeta potential, entrapment, and loading efficiencies were determined for TAC-NMF. An in-vitro drug release study of TAC-NMF, H^1^NMR analysis of TAC-NMF was performed. In-vitro cell cytotoxicity, in-vitro cellular uptake of TAC-NMF as compared to drug tacrolimus (TAC) were evaluated using various ocular cell lines. Finally, the in-vitro bioactivity of TAC-NMF was performed using sodium iodate induced inflammation and reactive oxidative stress on retinal pigment epithelial cells.

## 2. Materials and Methods

Tacrolimus was purchased from LC laboratories, Woburn, MA, USA. PEG-Hydrogenated castor oil-40 (HCO-40) was donated by Barnet and Octoxynol-40 (OC-40) was purchased from Rhodia Inc., Windsor, NJ, USA. For nanomicellar formulation preparation, double distilled deionized (DDI) water was used. HPLC-grade methylenedichlodide (DCM) and ethanol were purchased from Fisher Scientific (Pittsburgh, PA, USA). Fetal bovine serum (FBS) was purchased from Atlanta Biologics (Lawrenceville, GA, USA). TrypLE Express trypsin solution and Dulbecco’s modified Eagle’s medium (DMEM) were obtained from Invitrogen (Carlsbad, CA, USA). MTS reagent, CellTiter 96^®®^ Aqueous Non-Radioactive Cell Proliferation Assay was purchased from Promega, San Luis Obispo, CA, USA. Annexin V/FITC and PI-Staining solution was purchased from Molecular Probes^®®^ by Life Technologies, Hollow Road Madison, WI, USA. Fluorescein isothiocynate (FITC) reagent was purchased from Molecular Probes by Life Technologies^®®^, Oregon, Eugene, OR, USA. Dulbecco’s Phosphate-Buffered Saline (DPBS) was purchased from Gibco, Thermo Scientific Fisher. DCFDA/H_2_DCFDA—Cellular ROS Assay Kit was purchased from Abcam, Cambridge, MA, USA. Enzyme linked Immunosorbent Assay (ELISA) kits for TNF-α, IL-1β and IL-6 and human VEGF-A were purchased from e-Biosciences, San Diego, CA, USA. All other chemicals were of analytical reagent grade purchased from Thermo Fischer Scientific or Sigma-Aldrich (St. Louis, MO, USA) unless otherwise specified.

Human retinal pigment epithelium cell line D407 transfected with SV-40 adenovirus was kindly provided by Dr. Richard Hunt (University of South Carolina, Columbia, SC, USA). Monkey choroidal endothelial cell line RF/6A, human conjunctival cell line CCL20.2 and mouse macrophage RAW 264.7 cell line were purchased from the American Type Culture Collection (ATCC, Manassas, VA, USA). Cells were cultured in complete DMEM containing 10% heat inactivated FBS. The media contained high glucose and glutamine concentrations. The media also additionally contained 1% nonessential amino acids, 100 IU/mL penicillin and 100 IU/mL streptomycin. Cells were maintained at 37 °C in an incubator containing 5% CO_2_ and 90% relative humidity. Complete DMEM media was replaced every alternate day until the cells reached 80−90% confluence (2−3 days for D407, CCL 20.2 and 4–5 days for RF/6A cells).

### 2.1. Design of Experiment for Nanomicelles Preparation

A full factorial DOE was used to analyze various formulations having a fixed combination of independent variables. Three independent and five dependent variables were identified in the DOE. The independent variables were X1-sonication time (minutes), X2-HCO-40 (wt %) and X3-OC-40 (wt %). The dependent variables were Y1-hydrodynamic size (nm), Y2-polydispersity index (PDI), Y3-drug-entrapment efficiency (% *w*/*w*), Y4-drug-loading efficiency (% *w*/*w*) and Y5-zeta potential (mV). The full factorial design of experiments gave rise to 11 combinations or runs of independent variables for analyzing their effect on dependent variables. For each independent variable, higher (1), middle (0), and lower (−1) values were selected. HCO-40 and OC-40 amphiphilic polymers had the following values: HCO-40:1 = 3.5 wt %, 0 = 2 wt %, −1 = 0.5 wt % and OC-40:1 = 3.5 wt %, 0 = 2 wt %, −1 = 1 wt %. While sonication time had values like 1 = 25, 0 = 22.5 and −1 = 20 min. The final output of all the experiments is summarized in Table 1.

### 2.2. Preparation of Tacrolimus Nanomicellar Formulation

Tacrolimus nanomicellar formulation (TAC-NMF) 0.03% was prepared by solvent evaporation-film rehydration method of polymeric nanomicellar preparation as described by Mandal et al. [29]. Briefly, 3.0 mg of tacrolimus (TAC) and HCO-40 and OC-40 polymers (According to Table 1) were weighed and dissolved in 10.0 mL ethanol. The organic solvent was evaporated at a high speed under vacuum (Genevac, Ipswich, Suffolk, UK) to obtain a solid film. The resulting film was rehydrated with HPLC grade water. This was followed by filtration through a 0.22 µm nylon syringe filter (Figure 1). In a similar manner, placebo nanomicellar formulation (placebo NMF) was prepared, which excluded the step of TAC addition. TAC-NMF and placebo NMF were stored at 4 °C until further use.

#### 2.2.1. Formulation Characterization

Hydrodynamic size, polydispersity index (PDI) and the zeta potential of TAC-NMF and placebo NMF were determined by Dynamic Light Scattering (DLS) Zetasizer Nano ZS, Malvern Zetasizer, Westborough, MA, USA. For this, 700 µL of TAC-NMF or placebo NMF were placed in a glass cuvette in the DLS instrument (Malvern Zetasizer, Westborough, MA, USA). Transmission Electron Microscopy (TEM) and Scanning Electron Microscopy (SEM) was used to access the morphology of TAC-NMF. TAC-NMF in solution state was analyzed by TEM while lyophilized TAC-NMF was analyzed by SEM. Philips CM12 Scanning Transmission Electron Microscope was employed for SEM imaging and Philips XL30 ESEM-FEG Environmental Scanning Electron Microscope instrument was used for SEM imaging. TEM analysis of TAC-NMF was performed by placing 50 μL of TAC-NMF on a carbon-coated copper grid. Followed by negatively staining the sample by phosphotungstic acid. For SEM imaging, For the SEM analysis, lyophilized TAC-NMF was treated as a non-conductive wet sample and imagining was taken in the “wet” mode.

#### 2.2.2. Drug-Entrapment and Drug-Loading Efficiencies

HPLC method for determining TAC entrapped in nanomicelles: Concentration of TAC encapsulated within the core of the nanomicelles was determined by reverse phase Ultra-Fast Liquid Chromatography (RF-UFLC). A reverse phase C-18 column (Kinetex^®®^ 4.6µ, C18 100Å, 100 × 4.60 mm, Phenomenex In. USA) was used for the analysis of TAC entrapped in the nanomicellar formulation. The mobile phase was composed of a mixture of acetonitrile (ACN) and water (H2O) in 0.1% formic acid (FA). The flow rate was set at 0.3 mL/min, and the UVvis detector was set at a 220 nm wavelength. A standard curve of TAC drug was obtained by injecting varying concentrations of tacrolimus (500.0−1.95 µg/mL). The retention time of TAC was 5.4 min in a 10.0 min method.

Sample preparation: First, 1.5 mL of TAC-NMF was centrifuged at 20,000 rpm for 15 min. Then, 1 mL of the supernatant was collected. The supernatant was lyophilized to obtain a solid white residue containing polymers and drug TAC. One milliliter of dichloromethane (DCM) was added to the lyophilized product. Addition of DCM helps in the reversing the nanomicelles. DCM was evaporated under vacuum. This resulted in a solid pellet of the drug and reversed nanomicelles. The pellet was re-suspended using 1.0 mL of mobile phase. The entrapment and loading efficiencies of formulation F1–F11 prepared in triplicates was quantified by reverse phase HPLC and calculated using the following Equations:(1)$$\mathrm{Entrapment}\mathrm{Efficiency}=\frac{\mathrm{concentration}\mathrm{of}\mathrm{tacrolimus}\mathrm{in}\mathrm{TAC}-\mathrm{NMF}\mathrm{determined}\mathrm{by}\mathrm{RF}-\mathrm{UFLC}}{\mathrm{Concentration}\mathrm{of}\mathrm{tacrolimus}\mathrm{added}\mathrm{in}\mathrm{TAC}-\mathrm{NMF}}\times 100$$
(2)$$\mathrm{Loading}\mathrm{Efficiency}=\frac{\mathrm{concentration}\mathrm{of}\mathrm{tacrolimus}\mathrm{in}\mathrm{TAC}-\mathrm{NMF}\mathrm{determined}\mathrm{by}\mathrm{RF}-\mathrm{UFLC}}{\mathrm{Concentration}\mathrm{of}\mathrm{tacrolimus}\mathrm{added}\mathrm{in}\mathrm{TAC}-\mathrm{NMF}+\mathrm{concentration}\mathrm{of}\mathrm{polymers}\mathrm{added}}\times 100$$

### 2.3. Critical Micellar Concentration and Nanomicellar Viscosity Analysis

Two amphiphilic polymers were used in this study; HCO-40 and OC-40. These spontaneously form micelles in an aqueous medium above at certain concentration called the critical micellar concentration (CMC). The CMC of HCO-40, OC-40 and a mixture of HCO-40 and OC-40 (HCO-40: OC-40; 3.5:1) was determined using hydrophobic iodine as a probe. Nanomicellar formulations of varying concentrations of HCO-40 (7−6.51 × 10−9 wt %), OC-40 (2−3.72 × 10−9 wt %) and a mixture of HCO-40 and OC-40 (8−1.8 × 10−9 wt %) were prepared using serial dilutions. A solution of iodine (I_2_) and potassium iodine (KI) in the ratio I_2_: KI; 0.5:1 was prepared in deionized distilled water. Next, 100 µL of NMF was added to a 96-well plate. Then, 15 µL of I_2_:KI solution was added to each NMF. The 96-well plate was incubated for 5 h in dark at room temperature. The absorbance of hydrophobic iodine partitioned into the hydrophobic nanomicellar core was determined by microplate absorbance spectrophotometer (BioRad, Hercules, CA, USA). The absorbance was recorded at two emission wavelengths; 286 and 460 nm.

Viscosity of TAC-NMF was determined using an Ostwald–Cannon–Fenske viscometer. In short, the viscometer was filled with 10 mL of TAC-NMF using a pipette. Time taken for TAC-NMF to travel from Point A on the viscometer to Point B was noted. The same experiment was performed for DDI water. Viscosity of TAC-NMF was measured by the below equation:(3)$$\mathrm{Viscosity}\left(\mathrm{TAC}-\mathrm{NMF}\right)=\frac{\mathrm{Density}\left(\mathrm{TAC}-\mathrm{NMF}\right)\times \mathrm{time}\left(\mathrm{TAC}-\mathrm{NMF}\right)\times \mathrm{Viscosity}\left(\mathrm{water}\right)}{\mathrm{Density}\left(\mathrm{water}\right)\times \mathrm{time}\left(\mathrm{water}\right)}$$
Viscosity (water) = 0.89 centipoise (Cp) at 25 °C. Density (water) = 1 g/mL(4)

### 2.4. Nanomicellar Dilution Study

The effect of dilution on hydrodynamic size, zeta potential and PDI was determined using DLS (Zetasizer Nano ZS, Malvern Zetasizer, Westborough, MA, USA) Briefly, 0.03% TAC-NMF was diluted up to 200 times using HPLC grade water. Then, 700 µl of the resultant diluted TAC-NMF was analyzed for change in nanomicellar size.

### 2.5. ^1^H NMR Characterization

Proton nuclear magnetic resonance (^1^H NMR) spectroscopy (1H NMR and 13C NMR Varium-400 Mhz, Bruker Instruments, Durham, United Kingdom) was utilized to evaluate the entrapment efficiency of HCO-40 and OC-40 nanomicellar formulation to encapsulate TAC. TAC, TAC-NMF and placebo NMF were analyzed by Varian 400 MHz spectrometer NMR. Briefly, 1 mg/mL TAC-NMF (Formulation F6) and placebo NMF (Formulation F-6 without TAC) were lyophilized. Deuterated water (D_2_O) was used to re-suspend lyophilized TAC-NMF and placebo NMF. TAC (5 mg) and 1 mg/mL TAC-NMF (Formulation F6) lyophilized pellet was dissolved in CDCl_3_ and analyzed by ^1^H NMR.

### 2.6. In-Vitro Tacrolimus Dissolution and Drug Release

The in-vitro release of tacrolimus from TAC-NMF was determined by an UHPLC method. Briefly, 1.0 mL of TAC-NMF was transferred to a dialysis tubing of cut off molecular weight of 2000 Da. The dialysis tubing was surrounded externally by 5 mL buffer solution of 0.1% of Tween-20 in 1x PBS (PBST) in a 15 mL centrifuge tube. Additionally, 1.0 mL of TAC-NMF in a dialysis bag was suspended in stimulated tear fluid (STF). At every 24 h, 1.0 mL sample was collected and replaced with equal volume of fresh buffer solution to maintain sink conditions. The concentration of TAC released in PBST and STF buffer solutions was calculated using the UHPLC method for TAC stated above.

### 2.7. In-Vitro Cell Viability and Cytotoxicity Assay

#### 2.7.1. MTT Assay

Cellular cytotoxicity of TAC, TAC-NMF and blank NMF was determined on human retinal pigment epithelium (D407), human conjunctival (CCL20.2) and human retinal endothelial (RF/6A) cells. In-vitro cytotoxicity was determined by MTT assay. Cells were seeded 96-well plates at a cell density of 1 × 10^4^ cells/well and suspended in 200 µL of complete DMEM. Samples were prepared in serum free media (SFM) and filtered through a sterile 0.22 µm nylon filter. Triton-X 5% prepared in SFM served as the positive control and blank SFM without any treatment groups served as the negative control. The cell viability was calculated according to the formula:(5)$$\mathrm{Cell}\mathrm{Viability}=\frac{\mathrm{Absorbance}\mathrm{of}\mathrm{sample}-\mathrm{absorbance}\mathrm{of}\mathrm{negative}\mathrm{control}}{\mathrm{Absorbance}\mathrm{of}\mathrm{positive}\mathrm{control}-\mathrm{absorbance}\mathrm{of}\mathrm{negative}\mathrm{control}}\times 100$$

#### 2.7.2. Annexin V/FITC and PI-Staining

The effect of TAC-NMF, TAC and placebo NMF on cell apoptosis was determined by staining D407, CCL 20.2 and RF/6A ocular cells with Annexin V/FITC and propodium iodide (PI). The cells were seeded in 12-well plate at a cell density of 2 × 10^5^ cells/well. Following overnight attachment, the cells were treated with TAC, TAC-NMF and placebo NMF. The samples were prepared in SFM and filtered through a 0.22 µm filter in a biosafety cabinet to ensure sterility. Following 24 h of treatment, the cells were detached by addition of 200 µL of trypsin and washed twice with DPBS, and once with ice cold PBS, and collected by centrifugation in FACS tubes. The cells were finally stained with Annexin V/FITC and PI solution (Annexin V-FITC Apoptosis Staining Kit, Abcam) according to the manufacturer’s protocol. The cells were stored at 37 °C for 30 min in the dark before analysis by Fluorescence Assisted Cell Sorting (FACS).

### 2.8. In-Vitro Cellular Uptake Assay

The in-vitro cellular uptake of TAC-NMF was determined on D407, CCL 20.2 and RF/6A ocular cell lines with the help of FITC labelling.

#### 2.8.1. FITC Labelling

In-vitro uptake of TAC-NMF and TAC was determined by FITC labelling. OC-40 used in TAC-NMF formulation has a terminal –OH group available for conjugation. FITC was conjugated to OC-40 via EDC/NHS coupling reaction. Briefly, the optimized F-6 TAC-NMF was formulated in 50 mM PBS solution. This formulation was incubated with EDS/NHS (1:1) and FITC solution (1mg/mL in DMSO) at 4 °C in dark for 12 h. Similarly, for the naked drug TAC, EDS/NHS (1:1) and FITC solution (1 mg/mL in DMSO) was incubated in DMSO at 4 °C in dark for 12 h. The drug was dissolved in DMSO, since it was not soluble in 50 mM PBS. After 12 h, 1 mL of 50 mM NH_4_Cl was added to TAC-NMF-FITC and TAC-FITC solution to inactivate unreacted FITC. The final solution of FITC-labelled TAC-NMF and TAC was aliquoted and stored at −20°C until further use, as described in [29,33,34]. Figure 2 depicts the FITC conjugation reaction for TAC-NMF-FITC and TAC-FITC formation.

#### 2.8.2. In-Vitro Uptake Determination by FACS

Uptake of FITC-labelled TAC-NMF and FITC-labelled TAC in D407, CCL 20.2 and RF/6A was determined by acquiring samples by FACS. Briefly, cells were seeded in a 24-well plate with 5 × 10^4^ cells/well. Then, 20 µL of FITC-tagged TAC-NMF and FITC-tagged TAC was added to the respective wells for 3, 6, 9 and 12 h. At each time point, the cells were washed two times with DPBS (Gibco, Gaithersburg, MD, USA). Cells were collected in FACS tubes and centrifuged at 20,000 rpm for 5 min twice. The final sample was made in DPBS. The mean fluorescence intensity of TAC-FITC, TAC-NMF-FITC and control group was determined by FACS at an excitation wavelength of 490 nm.

#### 2.8.3. In-Vitro Uptake Determination by Confocal Microscopy

Cellular uptake of FITC-labelled TAC and FITC-labelled TAC-NMF was determined in D407 and CCL 20.2 cells. Labelling of FITC to TAC and TAC-NMF was carried out as described above. Both D407 and CCL 20.2 cells were seeded at a density of 1 × 10^4^ cells/well in an 8-chamber confocal microscopy slide (Nunc Lab-Tek 8 chambered, Thermo Fisher Scientific). Then, 10 µL of FITC-TAC (1 mg/mL) and FITC-TAC-NMF (1 mg/mL) was added to the respective chambers followed by termination of the treatment at 2 and 6 h. At each time point, cells were washed three times with DPBS, fixed with cold 4% buffered paraformaldehyde and stained with DAPI nuclear stain. The slides were covered with a coverslip and the sides were sealed. The cells were observed under Leica Confocal Laser Scanning Microscopy (Leica TCS SP5, Wetzlar, Germany).

### 2.9. In-Vitro Cellular Transport Determination

To determine transport of nanomicelles from the anterior segment of the eye to the posterior segment, an in-vitro model representing drug transport in the eye was developed. A transwell diffusion filter was utilized to determine the in-vitro permeability of TAC-FITC and TAC-NMF-FITC solutions. A Costar 12 well Transwell^®®^ Permeable Supports with a transparent filter of pore size of 0.4 µm was utilized for the transport studies. CCL 20.2 cells (cell count 3 × 10^4^ cells/well) were added to the top chamber. D407 cells were seeded in the bottom chamber with an average count of 5 × 10^4^ cells/well. After overnight incubation, D407 and CCL 20.2 cells were incubated for 3 days to achieve 100% confluence. Then, 20 µL of TAC-FITC and TAC-NMF-FITC was added on the upper chamber of the transwell inserts and the time-dependent uptake was seen in D407 cells from the bottom chamber. The mean fluorescence intensity was determined by FACS at excitation wavelength of 490 nm.

### 2.10. In-Vitro Biocompatibility Assay

#### 2.10.1. Pro-Inflammatory Cytokines in Macrophage Cells

A macrophage cell line, RAW 264.7 derived from mouse was used to determine the in-vitro biocompatibility of TAC-NMF and TAC using an Enzyme Linked Immunosorbent Assay (ELISA). Briefly, 2 × 10^4^ RAW 264.7 cells/well were seeded in a 96-well plate. The cells were treated with 20 µL of TAC-NMF and TAC solution 12 and 24 h. Following incubation, the supernatant medium of the cells was transferred into another 96 well plate which was further used for release of cytokines by ELISA. Here, bacterial lipopolysaccharide (LPS) derived from E. coli was used as a positive control and serum free medium was used as a negative control. LPS treatment on RAW 264.7 cells can induce transcription of genes, which regulate the production of proinflamatory cytokines [35]. ELISA kits for three proinflamatory cytokines—TNF-α, IL-1β and IL-6 (e-Biosciences, San Diego, CA, USA)—were utilized for evaluating the amount of pro-inflammatory cytokines released by the RAW 264.7 cells post-treatment.

#### 2.10.2. Apoptosis Assay in Kidney Cells

Madin-Darby canine kidney (MDCK) cells were used here to determine the effect of TAC, TAC-NMF and placebo NMF using Annexin V/FITC and PI-Staining. MDCK cells were seeded in 12-well plates at a cell density of 2 × 10^5^ cells/well in DMEM/F-12 media. The cells were treated with TAC, TAC-NMF and placebo NMF for 12 and 24 h. Following the treatment, cells were detached by addition of 200 µL of trypsin and washed twice with DPBS, and once with ice cold PBS, and collected by centrifugation in FACS tubes. The cells were finally stained with Annexin V/FITC and PI solution (according to the manufactures protocol) and stored at 37 °C for 30 min in dark before analysis by FACS.

### 2.11. Evaluation of Reactive Oxygen Species by DCFDA Assay

D407 cells were first incubated with 10 ug/mL of SI for 6 h in a 12-well plate at a cell density of 0.5 × 10^5^, followed by addition of TAC and TAC-NMF for 12 and 24 h. After treatment, cells were removed, washed twice with DPBS and centrifuged (20,000 r.p.m. for 5 min) twice. Finally, the cell suspension was treated with DCFDA dye for 30 min under dark. The cells’ fluorescence was analyzed by FACS.

### 2.12. In-Vitro Evaluation of TAC-NMF Bioactivity Using ELISA

Briefly, D407 cells 1 × 10^4^ cells were seeded in a 96-well plate. The cells were treated with 20 µL of 1 mg/mL solution of SI for 6 h. This was followed by addition of various treatment groups, such as 20 µL of TAC-NMF and TAC solution. These cells were evaluated for release of pro-inflammatory cytokines like TNF-α, IL-1β, IL-6 and VEGF-A in the supernatant medium at 12 and 24 h with the use of sandwich ELISA (e-Biosciences, San Diego, CA, USA) according to the manufacturer’s protocol.

### 2.13. Statistical Analysis

All the above experiments were conducted in triplicates or quadruplets. Results from three experiments which had similar results were used for further data analysis. All the results represented are mean of three experiments ± S.D. Student’s t test was used to determine statistical significance in the groups. A *p* value of less than 0.05 was considered as statistically significant and was indicated by (*). A *p* value of less than 0.01 was indicated as (**). Statistical analysis was performed by using Graph Pad Prism^®®^ 8.4.3 software (GraphPad Software, San Diego, CA, USA) for statistical analysis.

## 3. Results

### 3.1. Design of Experiment and Formulation Optimization

A DOE protocol was followed to study the effect of independent variables on dependent variables. The student version of JMP^®®^ 10.0 software was applied to for experimental design and data analysis. The independent variables were selected as (i) X1 sonication time (min), (ii) X2 HCO-40 (wt %) and (iii) X3 OC-40 (wt %). Each independent variable was assigned higher (+1), middle (0) and lower (−1) values. Table 1 shows the coded and uncoded design of independent variables for formulation preparation. On the other hand, formulation outcomes like (i) Y1 hydrodynamic nanomicellar size (nm), (ii) Y2 poly dispersity index (PDI), (iii) Y3 zeta potential (mV), (iv) Y4 entrapment efficiency (EE %) and (v) Y5 loading efficiency (LE %) were set as dependent variables. Nanomicellar formulations F1 to F11 were prepared by solvent evaporation-film rehydration method as described above. The results obtained from analyzing formulations F1-F11 were added to the full factorial design of experiment in the JMP^®®^ 10.0 software and a least square analysis of the independent variables on the dependent variables was performed (Table 2). For each dependent variable, Actual by Predicted Plot, Pareto chart, Surface Profiler and Prediction Equation were generated by the software for analyses of the interactions between the independent variables and dependent variables (Figure 2). The prediction equations for dependent variables are as follows:Y1 = 13.37 + 0.12 × X1 + 0.38 × X3 + 3.11 × X3 + X3 × (X2 × 0.33) + X1 × (X3−0.67) + X1 × (X3 × 0.39)(6)
Y2 = 0.34 + 0.8 × X1 + 0.06 × X2 + 0.092 × X3 + X1 × (X2 × 0.23) + X1 × (X3*0.36) + X3 × (X1 × 0.18)(7)
Y3 = 0.36 + 0.04 × X1 + 0.48 × X2 + 0.55 × X3 + X1 × (X2 − 0.37) + X1 × (X3 − 0.98) + X1 × (X3 − 1.35)(8)
Y4 = 66.97 + 0.27 × X1 + 15.86 × X2 + 10.87 × (X2−0.43) + X1 × (X3 + 0.78) + X2 × (X3 − 0.39)(9)
Y5 = 4.72 − 0.08 × X1 + 0.48 × X2 − 2.36 × X3 + X1 × (X2 × 0.029) + X1 × (X3 × 0.073) + X2 × (X3 × 0.16)(10)
where, Y1 = size, Y2 = PDI, Y3 = Zeta potential, Y4 = Loading efficiency, Y5 = Entrapment efficiency and X1 = sonication time, X2 = HCO-40 and X3 = OC-40.

The Actual by Predicted Plots provides a visual representation of how well the model fits and also compared the variation that occurs due to the dependent variables. It also gives a correlation between the results obtained from experimentations and the outcomes predicted by the software [36]. Actual by Predicted Plot for Size, PDI, Zeta Potential, Entrapment Efficiency and Loading Efficiency have a probability (P) values of 0.989, 0.9507, 0.222, 0.018 and 0.1042, respectively [Figure 2(A(i),B(i)),C(i),D(i),E(i))], which indicate a positive correlation between calculated and predicted values. The least square fit model runs, also developed Pareto charts for each dependent variable to understand the interaction between which independent variable is the most significant. The Pareto bars crossing or touching the blue lines indicate factors reaching statistical significance (α = 0.05). For size, two factors interaction between HC0-40 and OC-40 touched the blue line indicating statistically significant effect on nanomicellar size [Figure 3(A(iii))]. On the other hand, factors like sonication time, sonication time × HC0-40, HCO-40, sonication time × OC-40 and OC-40 [Figure 3(A(iii),B(iii),C(iii),D(i),E(iii))] did not show a statistically significant effect influencing TAC nanomicellar size. For PDI, two-factor interaction between sonication time and OC-40 touched the blue line indicating statistically significant effect on PDI [Figure 3(B(iii))]. For Zeta Potential, the bar for single factor OC-40 touched the blue line indicating statistically significant effect on it [Figure 3(C(iii))]. For entrapment and Loading efficiencies, single term HCO-40 and dual effect of HCO-40 and OC-40, respectively, show statistical significance [Figure 3D,E(iii)].

Contour plots for size, PDI, zeta potential, loading and entrapment efficiency [Figure 3(A(ii),B(ii),C(ii),D(ii),E(ii))] were respectively developed. These are three-dimensional representation of changes in dependent variables on shifting independent variables. When the independent input variables were set to sonication time (20 min), HCO-40 (3.5 wt %) and OC-40 (1 wt %), a lower size, PDI, zeta potential and higher loading efficiency and high entrapment efficiency were predicted. Along with above analysis of relationship between independent and dependent variables, a prediction profiler for the variables was established. Prediction profiler could help in adjusting the levels of independent variables in specific combinations, where the values of dependent variables can be predicted.

Before developing the prediction profiler (Figure 4), independent factors such as size, PDI and zeta potential were set to minimum desirability and loading and entrapment efficiencies were set to maximum desirability. Input variables sonication time (20 min), HCO-40 (3.5 wt %) and OC-40 (1 wt %) resulted in lower nanomicellar size (13.01 nm), lower PDI (0.34), lower zeta potential (1.601), higher loading efficiency (0.353%) and higher entrapment efficiency (99.77%). The simulations obtained from the prediction profiler were in good agreement with the results acquired by actual experimentation specifically for Formulation-6 (F6). Hence, the specific combination of independent variables in F6 in Table 2 was used for further experimentation. Experiment simulation results by JMP 10.0 software and results obtained from the actual experiments suggest that polymer concentration (HCO-40 and OC-40) and sonication time when kept at a certain levels can yield a TAC-NMF with low size, PDI, zeta potential, high loading and high entrapment efficiencies. Hence, F-6, which showed optimization of all the dependent outcomes in the simulation studies and in the actual experiment performed, was selected as the optimized formulation for further studies.

### 3.2. Formulation Characterization

#### 3.2.1. Design of Experiment, Size, PDI and Zeta Potential

A design of experiment (DOE) protocol was followed to study the effect of independent variables on dependent variables. The student version of JMP^®®^ 10.0 software was employed for the experimental design. The independent variables were selected as (i) X1 sonication time (min), (ii) X2 HCO-40 (wt %) and (iii) X3 OC-40 (wt %). Nanomicellar formulations F1 to F11 were prepared by solvent evaporation-film rehydration method as described above. The results obtained from analyzing formulations F1-F11 were added to the full factorial design of experiment in the JMP^®®^ 10.0 software and a least square analysis of the independent variables on the dependent variables was performed.

Amphiphilic polymers in aqueous solution above CMC form spherical nanomicelles, with a hydrophobic core and a hydrophilic corona facing the exterior aqueous environment. Such nanomicelles can entrap hydrophobic drugs in their core and serve as an excellent drug delivery system for the same and greatly improving its solubility [2]. Here we prepared nanomicellar formulations of TAC using a mixture of amphiphilic polymers such as HCO-40 and OC-40 (Figure 1A). Solvent evaporation-film rehydration method was employed for TAC-NMF preparation. The solvent and drug film were rehydrated using HPLC grade water (Figure 1B). Various TAC-NMF prepared (F1-F11) had a hydrodynamic size range of 13.34–22.72 nm, PDI of 0.16–0.38 and zeta potential of 0.512–1.896 mV. The optimized formulation F-6 (Table 2) had the lowest size, PDI, zeta potential and maximum entrapment and loading efficiencies. Nanomicellar formulation F-6 had the highest entrapment efficiency compared to all other formulations. An entrapment efficiency of around 97% can indicate complete entrapment of the hydrophobic drug TAC. This can imply that the drug precipitation during storage can be reduced. Along with this, F-6 has the smallest size of all the other nanomicellar formulations. This can indicate stronger interaction between the polymers and the drug. Nanomicelles with smaller size (<100 nm) and neutral to positive charge can allow better absorption into ocular tissues and allow better tissue penetration.

In addition to enhanced penetration, nanomicelles with smaller size can also be transported through the conjunctival-scleral pathway. The human sclera has aqueous pores or channels having an internal diameter from 20–80 nm [37]. This enables the transport of nano-sized drug carriers such as nanomicelles to the retinal tissue. [38]. Nanomicelles with smaller size can form clear solutions which serve as an excellent drug delivery vehicle for ophthalmic drugs. [28]. The optimized formulation F-6 had a hydrodynamic size of 15.41 nm (Figure 5A), a PDI of 0.25, and a zeta potential 0.512 mV (Figure 5B). TAC-NMF is clear solution of tacrolimus and its clarity can be compared to DDI water (Figure 5C). A drug free mixed nanomicellar formulation of HCO-40+OC-40 (HCO-40: OC-40: 3.5:1 wt %) was also prepared by the same method as stated above. The drug free or placebo NMF had size of 17.36 nm, PDI of 0.263 and zeta potential 0.36 mV. The larger size of the drug free nanomicelles vs. the smaller size of the drug-loaded nanomicelles can be attributed to higher hydrophobic interactions in the drug-loaded nanomicelles. TAC being a hydrophobic drug strongly interacts with the hydrophobic portion of the amphiphilic polymers: HCO-40 and OC-40. This results in the shrinking of the core of the nanomicelles and thus lowers the size. Such hydrophobic interactions are present in placebo nanomicelles. TEM and SEM analysis of TAC-NMF revealed spherical nanomicelles (Figure 5D,E).

#### 3.2.2. Entrapment and Loading Efficiencies

Tacrolimus entrapped in TAC-NMF was quantified by a direct quantification method using the reverse micellization process of reverse engineering. The drug release was quantified by RP-UFLC using the method described above. Depending on the variations in the total amphiphilic polymer (HCO-40 + OC-40), the entrapment efficiency varied from 72.71 to 97.13%, and the loading efficiency varied from 0.55 to 10.9% in F1-F11 (Table 2). Among formulations F1–F11, F-6 exhibited highest entrapment efficiency, and F-8 exhibited highest loading efficiency. The higher loading efficiency for formulation F-8 can be attributed to lower polymer concentration as compared to other formulations. It can be noted here that although the amount of drug TAC added to the formulations was the same (0.3 mg/mL), the formulations exhibit varying loading and entrapment efficiencies due to the difference in the polymer concentration and sonication time.

### 3.3. Critical Micellar Concentration and Nanomicellar Viscosity Analysis

Critical micellar concentration (CMC) is an important factor determining the performance of the drug delivery carrier, which can ultimately affect the drug bioavailability [39]. A higher value of CMC for a given polymer can result in premature release of the drug in ophthalmic applications. Static ocular barriers such as the tear film barrier, corneal barrier and dynamic ocular barriers like blood aqueous barrier, vitreal barrier and blood retinal barrier can result in the destruction of the nanomicellar structure. This can result in premature drug release at a non-target area and loss of topically and intravitreally applied ocular formulations [40]. This loss can be ameliorated by encapsulating hydrophobic drugs in the core of nanomicelles with lower CMC [41]. A lower CMC value for a nanomicellar formulation can also help ion prevention of the disruption of the nanomicellar structure on tear dilution. Ophthalmic formulations when installed in the precorneal pocket are rapidly diluted by the tears and circulating tear fluid turn over [42,43]. In this study, CMC was determined for HCO-40, OC-40 and a mixture of HCO-40 and OC-40 in the F-6 optimized formulation ratio; 3.5:1.0 wt %. The purpose of this study was not only to calculate the CMC of the polymers, but also to determine whether mixture of the polymers had any effect on the total CMC value. The CMC of OC-40, HCO-40 and mixture HCO-4+OC-40 (ratio 3.5:1) was 0.0685 wt %, 0.0447 wt % and 0.0349 wt %, respectively. A lower CMC value of the mixture of the polymers indicated that polymeric mixture of HCO-40 and OC-40 could be more stable as compared to the individual polymers.

The viscosity of liquid formulations can be an important property to understand their syringeability and flow properties. Especially for ophthalmic formulations, viscosity determination can be essential, as TAC-NMF can be used either topically or as intravitreal injection. For topically instilled formulations, viscosity plays an important role in the mean residence time of the formulation and its drainage. Similarly, ophthalmic formulations administered intravitreally should have a viscosity between that of water and vitreal humor fluid [44]. The viscosity of the optimized F-6 formulation was 1.12 ± 0.05 cP (mean ± SD) (*n* = 3), at RT. This value was almost near to the viscosity value for water, which is 0.89 cP.

### 3.4. Nanomicellar Dilution Study

Human ocular anatomy consists of various dynamic and static ocular barriers to protect the eye from various xenobiotics. Such barriers also retard the passive absorption of topically administered therapeutic agents and drug loaded nanocarriers. Amongst dynamic ocular barriers, tear turnover and the recycling of aqueous and vitreous humor are the major factors concerning the bioavailability of ophthalmic formulations. Ophthalmic formulations including eye drops are administered into the precorneal pocket. The precorneal pocket can hold topical ophthalmic formulation up to 10 µL and the tear fluid turn over occurs at a rate of 0.7 μL/min. This implies that once the topical formulation is dropped into the precorneal pocket, it is diluted by the tears. Hence, the tear dilution study for TAC-NMF was performed keeping such physiological mechanism in mind. Ophthalmic formulations administered intravitreally also undergo dilution in the vitreous humor. TAC-NMF was diluted up to 200 times its original volume and the resultant size, PDI and zeta potential were measured. The results are shown in Table 3. After 200 times dilution, the nanomicellar size increased from 15.36 nm to 17.01 nm, PDI increased from 0.183 to 0.672 and zeta potential from 0.522 mV to 1.93 mV. This suggests indicated that dilution up to 200 times had very little effect on the nanomicellar size and significant change in PDI and zeta potential. Similar results were obtained in a previously reported study, where similar polymers were used to enhance ocular drug delivery of triamcinolone acetonide and curcumin [30,44].

### 3.5. ^1^H NMR Characterization

^1^H NMR analysis of TAC-NMF, placebo NMF and TAC was performed to determine the entrapment of tacrolimus drug in the core of TAC-NMF. For this purpose, TAC-NMF, placebo NMF were lyophilized and dissolved in D_2_O and TAC was dissolved in deuterated chloroform (CDCL_3_). Lyophilized TAC-NMF was also dissolved in CDCL_3_ so that the drug in the core of the nanomicelles could be released in the solution for quantification. Figure 6 depicts the ^1^H NMR spectra for TAC-NMF in CDCL_3,_ TAC-NMF and blank-NMF in D_2_O and TAC in CDCL_3_. The spectra for TAC in CDCL_3_ are in accordance with previously published results [45]. A sharp peak for oxyethylene group -CH_2_-CH_2_-O (δ = 3.8 ppm), hydroxyl protons (PEG) (δ = 4.5 ppm), and moderate signals for methylene =CH_2_ (δ = 1.3 ppm) and methyl –CH_3_ (δ = 0.7 ppm) groups of HCO-40 amphiphilic surfactant polymer were recorded. The aromatic protons of OC-40 showed a weak signal at δ = 6.8 and δ = 7.2 ppm. When the spectra for blank nanomicelles and TAC-NMF in D_2_O were compared, a similar set of peaks for the ^1^H NMR spectrum was observed. In addition, no peaks for TAC were observed in TAC-NMF in D_2_O. This absence for signal for TAC indicated and confirmed the complete entrapment of drug tacrolimus in the nanomicellar formulation. When TAC-NMF was dissolved in CDCL_3,_ the peaks for polymers HCO-40 and OC-40 and TAC were observed. The peak of TAC in TAC-NMF in CDCL_3_ was relatively small because of the lower abundance of TAC compared to the polymers used in the nanomicellar formulation. This confirmed that TAC was encapsulated within the core of the nanomicelles.

### 3.6. In-Vitro Dissolution and Drug Release

In-vitro drug release studies are pre-requisite to find out the time line of a formulation to release the active pharmaceutical agent. We performed this study with sufficiently diluted solutions, where the drug release was not limited to its solubility. In-vitro release studies reflect the kinetic profile of drug release and predict the performance of the nanocarrier system in vivo. In-vitro drug release tests performed in PBS plus tween-20 (PBST), which simulate similar features in-vivo systems. In addition, sampling from the external buffer fluid at regular intervals allows maintaining adequate sink conditions as found in-vivo system [46]. To evaluate the in-vitro drug release behavior of TAC from TAC-NMF, we performed in-vitro drug release test of TAC-NMF in PBST and STF with constant shaking at 37 °C. As evident from the Figure 7, TAC was rereleased from TAC-NMF in a controlled and sustained manner for a period of more than 20 days. Up to 100% release of TAC was obtained at day 22. In STF, 95% of TAC release was obtained from TAC-NMF in 18 days. These results indicate TAC release in STF is different as compared to PBST. This difference can be due the varying solute concentration and pH value of the two formulations. Hence, this suggests that TAC-NMF can sustain release TAC when applied as an ophthalmic formulation due to the controlled drug release. In-vitro drug release can give an idea about the in vivo behavior of the nanocarrier in the aqueous humor, vitreous humor and tear fluids.

### 3.7. In-Vitro Viability and Cytotoxicity Assay

In-vitro cytotoxicity of TAC, TAC-NMF and placebo NMF was evaluated on D407, CCL 20.2 and RF/6A ocular cells. This study was performed to assess the safety of TAC-NMF and its components on the ocular cells. Post-treatment with TAC, TAC-NMF and placebo NMF, the cells were analyzed for cell viability using molecular biology assays like MTT assay, PI-Staining and Annexin V-FITC Staining assay. Furthermore, in-vitro cytotoxicity was assessed by assessed for the lactose dehydrogenase released in the serum by the cells by LDH assay (Appendix A).

#### 3.7.1. MTT Assay

Initially placebo NMF was tested on the three cell lines for 24 h. Figure 8A depicts the cell viability (%) of D407, CCL 20.2 and RF/6A cells when treated with increasing concentrations of placebo NMF (20–40 mg/mL of HC0-40 + OC-40 in the ratio HCO-40:OC-40: 3.5:1). D407 and CCL 20.2 cells depict almost 90% cell viability for most of the placebo NMF formulations. RF/6A cells, on the other hand, show slightly lower cell viability (85–92%) as compared to D407 cells and RF/6A cells. This demonstrates the nontoxic effect of the placebo NMF on D407 and CCL 20.2 cells. The cytotoxic effect of TAC drug alone and TAC-NMF was studied on D407 (Figure 8B), CCL 20.2 (Figure 8C) and RF/6A cells (Figure 8D). Cells were treated with TAC and TAC-NMF (tacrolimus TAC concentrations: 100–1000 µg/mL) for 24 h, and then the cell viability was measured by MTT assay. As can be seen in Figure 8B–D, cell viability (%) is statistically higher in the TAC-NMF group as compared to the TAC group for all TAC concentrations. This suggests the cytotoxicity of the drug can be reduced by encapsulating potent drugs like TAC in a nanomicellar formulation. Controlled release of the drug from the core of the nanomicelles and reduced concentration can lower the cytotoxicity of highly potent drugs such as TAC. In addition, sustained supply of drug for a prolonged period may be better for controlling inflammation.

#### 3.7.2. Annexin V/FITC and PI-Staining

Apoptosis induced by placebo NMF, TAC and TAC-NMF formulation in D407, CCL 20.2 and RF/6A ocular cells was determined by Annexin V/FITC and Propodium Iodide (PI) staining. The cells were treated for 24 h with placebo NMF, TAC and TAC-NMF formulations. Placebo NMF contained the final optimized formulation without the drug tacrolimus dissolved in SFM. TAC and TAC-NMF formulations, Figure 9A depicts the percentage of cells having undergone apoptosis after treatment with placebo NMF, TAC and TAC-NMF formulations in D407, CCL 20.2 and RF/6A ocular cells. Annexin V/FITC and PI-Staining helps to detect late stage apoptotic cells and necrotic cells. Figure 9A depicts the effect of placebo NMF on D407, CCL 20.2 and RF/6A cells. It can be seen that the percentage of apoptotic cells in the control group and the treatment group are almost comparable and similar. This can indicate that the placebo NMF does not cause cell death largely. Figure 9B represents the effect of TAC and TAC-NMF at various concentrations on D407, CCL 20.2 and RF/6A cells. As seen in these figures, the percentage of cells undergone apoptosis is higher for the TAC treatment group at higher concentrations as compared to the control group in all the three cell lines. Interestingly, this percentage is lower in all the cell lines with TAC-NMF treatment group. This statistically lower percentage of apoptotic ocular cells in the TAC-NMF group can be an indication that the controlled release of the drug can responsible for making tacrolimus drug less toxic to the ocular cells.

### 3.8. In-Vitro Cellular Uptake and Bio-Distribution of TAC-NMF

The in-vitro cellular uptake of TAC-NMF and TAC was determined in multiple ocular cell lines to understand the localization of the nanomicellar formulation within the cells. This study also helped to compare the drug absorption in nanomicellar formulation and the drug alone. Two specific approaches were utilized to study in-vitro cellular uptake of TAC-NMF. A quantitative analysis involved the detection of fluorescent cells using FACS. Here, ocular cells, D407, CCL 20.2 and RF/6A were incubated with FITC-labelled TAC-NMF and FITC-labelled TAC solutions.

#### 3.8.1. FACS Analysis

TAC-NMF and TAC were labelled with FITC as mentioned in the procedure above. D407, CCL 20.2 and RF/6A ocular cells were harvested and seeded in a 24-well plate. Then, 20 µg/mL of either FITC-labelled TAC-NMF or FITC-labelled TAC solution was added to the wells and incubated at 37 °C for 3, 6, 9, and 12 h. A time-dependent in-vitro uptake of FITC-labelled TAC-NMF was determined. The results of the assay are shown in Figure 10. The mean fluorescence intensity of FITC-labelled TAC-NMF was statistically higher than the corresponding control and FITC-labelled TAC groups in 6, 9 and 12 h treatment groups in all the cell lines. In the three-hour treatment group, the mean fluorescence intensity of FITC-labelled TAC-NMF was statistically higher than the FITC-labelled TAC for CCL 20.2 and RF/6A cells. The relatively lower fluorescence intensity in D407 cells as compared to CCL 20.2 and RF/6A cells in the three hour treatment group can be explained by the fact that D407 cells express a myriad of tight junction proteins which help in the formation of the outer blood–retinal barrier [47]. D407 cells contain tight junction proteins like zonula occludens and claudins [48]. Such tight junction proteins can lower the permeation of nanocarriers and various therapeutic agents into these cells [49,50].

#### 3.8.2. In-Vitro Bio-Distribution of TAC-NMF Using Confocal Laser Scanning Microscopy Analysis

In-vitro cellular uptake of FITC-labelled TAC-NMF was performed using Confocal Laser Scanning Microscopy (CLSM) for the visual representation and qualitative analysis of TAC-NMF as compared to TAC. D407 and CCL 20.2 were incubated with either FITC-labelled TAC-NMF or FITC-labelled TAC for 2 or 6 h. The cells were washed, fixed and mounted for observation using CLSM. As seen in Figure 11, the fluorescence at the two-hour time point was higher in the TAC-NMF group as compared to TAC in D407 and CCL 20.2 cells. Similar results were obtained in both the cell lines for 6 h. The internalization of TAC-NMF in these cells increased with time. On the other hand, a weaker fluorescence for FITC-labelled TAC was seen in the entire cell lines at all the time points as compared to the FITC-labelled TAC-NMF group. This implies that the nanomicellar formulation internalizes in the ocular cells at a faster rate than the corresponding drugs. This can be attributed to the stronger interaction between the amphiphilic polymers and the lipid bilayer of the cells. In addition, when the fluorescence intensity between the two cell lines is compared, CCL 20.2 exhibits a relatively higher fluorescence at time point 2 h as compared D407 cells. This can be due the presence of tight junction proteins [10,11]. The results obtained from the CLSM analysis of TAC-NMF as compared to TAC can be correlated to the results obtained for the FACS analysis.

### 3.9. In-Vitro Cellular Transport of TAC-NMF

Although in-vitro cellular uptake studies of FITC-labelled TAC-NMF using FACS and CLSM indicated the statistically significant uptake of the nanomicellar formulation in D407, CCL 20.2 and RF/6A cells, it is important to verify the in-vitro cellular transportation potential of TAC-NMF. For this purpose, a dual chamber model was established which had CCL 20.2 cells growing on the upper chamber and D407 cells growing on the bottom chamber. This dual chamber model represented an in-vivo condition for the human eye. This in-vitro model was developed to evaluate the transport of nanomicelles from a layer of CCL 20.2 cells and their subsequent up take by D407 cells in the in the bottom chamber. Various treatment groups like FITC-labelled TAC-NMF and FITC-labelled TAC were added to the upper chamber, and time-dependent uptake was quantified in the bottom chamber cells by FACS. There was no significant difference observed in the transport of TAC-NMF at 3 and 6 h (Figure 12), but at 9 and 12 h, the transport of TAC-NMF was significantly higher than TAC group. TAC-NMF crossed the CCL 20.2 cell monolayer by paracellular route, which enabled its active absorption into D407 cells in the bottom chamber [51,52,53]. This can suggest that a nanomicellar system consisting of a mixture of amphiphilic polymers such as HCO-40 and OC-40 can be useful for the ophthalmic delivery of hydrophobic drugs such as tacrolimus to the back of the eye via the transscleral route. In addition, a lower CMC and controlled release of tacrolimus from the nanomicellar core can aid the effective delivery of TAC-NMF to the back of the eye via the conjunctival- scleral pathway for retinal disorders

### 3.10. In-Vitro Biocompatibility Assay

The in-vitro biocompatibility of placebo NMF, TAC-NMF and TAC was determined in macrophage cells and kidney cells. RAW 246.7 macrophage cells and MDCK kidney cells were used to establish the biocompatibility of various treatment groups. RAW 246.7 cells are widely used as a model to assess the inflammatory response and release of inflammatory cytokines after treatment with various ocular nanomicellar formulations by ELISA [28]. Along with the analysis of inflammatory cytokines on the macrophage cells, the effect of TAC, TAC-NMF and placebo NMF was also observed on MDCK kidney cells. This was done to assess the effect of tacrolimus on the kidney cells, as tacrolimus is a nephrotoxic drug. Although the systemic bioavailability of TAC-NMF administered intravitreally to the eye will have negligible systemic effects, its effect on MDCK cells was evaluated using Annexin V/FITC and PI-Staining.

#### 3.10.1. Pro-Inflammatory Cytokines in Macrophage Cells

RAW 246.7 macrophage cells were treated with placebo NMF, TAC and TAC-NMF for 12 and 24 h. This was followed by analyzing the extracellular release of pro-inflammatory cytokines such as TNF-α, IL-6, and IL-1β using sandwich ELISA. The results of this study are shown in Figure 13. It is seen that TNF-α, IL-6, and IL-1β release from the cells after 12 and 24 h of treatment was comparable to the negative control. Specifically in Figure 13A, TNF-α concentration was higher for placebo NMF at 12 and 24 h as compared to the negative control, but TAC and TAC-NMF had similar values as compared to the negative control. This can indicate that tacrolimus can inhibit the release of TNF-α from RAW 246.7 cells. Similar results can also be found for Figure 13B, which compares IL-6 concentration released from the cells by the various treatment groups at 12 and 24 h. Here it can be noted that the release of IL-6 at 24 h in some treatment groups, such as TAC and TAC-NMF, at 24 h was statistically lower than at 12 h. A similar trend as stated above can be seen for IL-1β in Figure 13C. Such results can be an indication that polymeric TAC-NMF can be regarded as safe for further ophthalmic applications.

#### 3.10.2. Apoptosis Assay in Kidney Cells

MDCK kidney cells were used to determine the effect of placebo NMF, TAC and TAC-NMF formulations. This was done to evaluate if TAC-NMF can have lower toxicity and induce apoptosis to a lower percentage than the naked drug. In addition, this assay was performed to determine if the placebo NMF has any cytotoxicity on the MDCK kidney cells. Here the cells were treated with placebo NMF, TAC and TAC-NMF for 12 and 24 h following staining with Annexin V/FITC and PI, and subsequent analysis by FACS. Figure 14 presents an in-vitro biocompatibility study of TAC and TAC-NMF in MDCK kidney cells by determining by Annexin V/FITC and PI-Staining at 12 h (**A**), 24 h (**B**), and the effect of placebo NMF (**C**) at 12 and 24 h. It can be observed in Figure 14A that as compared to the TAC group, the percentage of apoptotic cells in the TAC-NMF group (0.3, 0.5 and 1.0 mg/mL groups) increased when treated for 12 h. Similar results can also be seen for MDCK cells treated with TAC and TAC-NMF formulations for 24 h (Figure 14B). The percentage of apoptosis in the TAC-NMF group is comparable to the control group and is also statistically lower than that for TAC treatment group when treated for 24 h. Interestingly, the placebo NMF did not induce higher percentage of apoptosis in MDCK cells as compared to the control group (Figure 14C). These results can be an indication that TAC-NMF has lower nephrotoxic potential than TAC drug alone on MDCK kidney cells, primarily because of the controlled release behavior of the nanomicellar formulation.

### 3.11. Evaluation of Reactive Oxygen Species by DCFDA Assay

The ability of TAC-NMF and TAC to lower the Reactive Oxygen Species (ROS) in D407 cells was evaluated by 2′,7′-dichlorodihydrofluorescein diacetate (H_2_DCFDA) assay. Sodium Iodate (SI) is an agent which causes oxidative stress in the retinal pigment epithelial cells and can be used in in-vitro [54] and in-vivo models for early age-related macular degeneration [55,56]. Reactive Oxygen Species (ROS) produced in D407 cells pretreated with SI was determined after the treatment of TAC (0.1, 0.3, 0.5, 1.0 ug/mL) and TAC-NMF (0.1, 0.3, 0.5, 1.0 ug/mL) at 12 and 24 h. This was performed with the help of DCFDA Assay and the ROS present in D407 cells were quantified by FACS. Wet AMD constitutes an increase in oxidative stress and inflammation in the retinal pigment epithelium. In this study, we employed the use of SI to induce oxidative stress to the retinal pigment epithelium cells; D407 cells. SI is considered to be a retinotoxin because of its ability to induce oxidative stress in the retinal pigment epithelial cells [57]. SI-induced oxidative stress in mice is commonly used as an experimental model for the study of therapeutic agents for wet AMD. Here, we hypothesized that SI can also induce oxidative stress in D407 and can be used as an in-vitro model for early AMD. Figure 15 depicts the mean fluorescence intensity of D407 cells pretreated with SI for 6 h and the treated with TAC or TAC-NMF for 12 and 24 h. It can be seen that higher concentrations of TAC and TAC NNF can significantly reduce the ROS produced by SI pretreated D407 cells. The ROS produced by D407 cells when treated TAC 1.0 ug/mL at 12 and 24 h is similar to the control. The same is also true for TAC-NMF at 1.0 ug/mL treated D407 cells. This can indicate that TAC and TAC-NMF can reduce the ROS in SI pretreated D407 retinal pigment epithelial cells.

### 3.12. In-Vitro Evaluation of TAC-NMF Bioactivity Using ELISA

The in-vitro bioactivity of TAC-NMF, TAC and placebo NMF was evaluated on D407 cells pretreated with sodium iodate (SI). SI induces oxidative stress in the RPE cells, resulting in photoreceptor apoptosis and early stage AMD [58,59]. An early stage of AMD constitutes an increase in oxidative stress and inflammation in the retinal pigment epithelium. In this study, we employed the use of SI to induce oxidative stress to the retinal pigment epithelium cells; D407 cells [57,60]. Here we postulated that SI can induce oxidative stress in D407 and can be used as an in-vitro model for early AMD. For this purpose, D407 cells were seeded in 96-well plates, and then treated with SI for 6 h followed by treatment with either placebo NMF, TAC and TAC-NMF for 12 and 24 h. Inflammatory cytokines like TNF-α, IL-6, IL-1β and VEGF-A released in the supernatant medium were evaluated using sandwich ELISA. As can be seen in Figure 16A–D, TNF-α, IL-6, IL-1β and VEGF-A for D407 cells treated with SI were higher than the control. This can aid in validating the model suitable for inducing oxidative stress and subsequent release of inflammatory cytokines, and VEGF-A. Similarly, in all the five graphs, the number of inflammatory cytokines and VEGF-A released by placebo NMF was slightly less than that of the SI-treated group. However, the amount of inflammatory cytokines and VEGF-A released by TAC-NMF and TAC was significantly lower in a concentration-dependent manner as compared to the drug-free NMF and SI-treated cells groups. This suggests a concentration-dependent lowering of TNF-α, IL-6, IL-1β and VEGF-A in D407 cells after the treatment of TAC and TAC-NMF for 24 h.

## 4. Discussion

Self-assembling, optically clear, aqueous nanomicellar solution of tacrolimus was prepared by the solvent evaporation-film rehydration method using an experimentally determined blend of polymers HCO-40 and OC-40. Previous studies with the polymers demonstrated that a mixture of HCO-40 and OC-40 gave rise to nanomicelles with a size less than 40 nm that were suitable for ocular drug delivery [29,30,31]. Terminal PEG group of HCO-40 can impart stealth properties to the nanomicelles and increase their stability. The low molecular weight of HC0-40 and the surfactant nature of OC-40 helps in the formation of nanomicelles with a lower size. HCO-40 and OC-40 polymers GRAS polymers approved by the FDA. They are also the constituent of Cequa^®®^ nanomicellar formulation approved for dry eye disorder [28]. TAC-NMF formulation was optimized by JMP^®®^ DOE software using full factorial design of experiment. Experimental analysis and software simulations demonstrated that a specific mixture of amphiphilic polymers, HCO-40 and OC-40 in the concentration of 3.5 wt % and 1.0 wt % demonstrated lower size, lower PDI and near zero zeta potential. Lower size of the nanomicelles helps in rapid absorption in the ocular tissues by receptor mediated endocytosis or passive diffusion and absorption. A lower PDI suggested that the TAC-NMF was monodisperse and has a uniform size distribution. A near zero zeta potential helps in the absorption of TAC-NMF into the cell cytoplasm. There is lower repulsion between the cell membrane and the TAC-NMF corona [28]. The optimized formulation demonstrated high entrapment and loading efficiencies. This can indicate that the nanomicelles could completely entrap hydrophobic drug, tacrolimus. This caused less drug wastage and lower drug precipitation imparting higher stability to the formulation. In addition, The CMC of the mixture of polymers was lower than the individual polymers. This justifies the use of a combination of the polymers HCO-40 and OC-40. A lower CMC indicates stable nanomicelles upon dilution. This was seen when TAC-NMF was diluted 200 times. TAC-NMF dilution only marginally increased the nanomicellar size, PDI and zeta potential. This can indicate the TAC-NMF stability in aqueous and vitreous humor. H^1^NMR analysis of TAC-NMF denoted almost complete encapsulation of TAC in the core of the nanomicelles. The in-vitro release study of TAC-NMF demonstrated a steady release of TAC over a period of 22 days. Cequa^®®^, a nanomicellar formulation approved for dry eye disorder, also has a blend of HCO-40 and OC-40 polymers [28]

In-vitro cytotoxicity studies of placebo NMF depicted minimal toxicity on ocular cell lines, while TAC and TAC-NMF depicted dose dependent toxicity. In addition, it was interesting to note here that the cell viability was higher for TAC-NMF as compared to TAC, because of the slower release of the drug. Similarly, the TAC-NMF induced a lesser percentage of apoptosis in the ocular cells as compared to TAC drug alone at 12 and 24 h. Cellular uptake studies for TAC and TAC-NMF demonstrated an increased uptake in a time dependent manner. The results were consistent with a previously published study by Mandal et al. which discussed the cellular uptake of cidofovir prodrug nanomicelles in retinal and cornel cells [29]. An intracellular distribution study for the TAC-NMF depicted efficient uptake and distribution of the nanocarrier in retinal D407 and conjunctival CCL 20.2 cells at 6 h. This can implement that the HCO-40 and OC-40 nanomicellar system can not only be an efficient carrier system for hydrophobic drugs, but can also integrate well with the cell membrane for effective intracellular drug delivery [28,29]. The duel-chamber uptake study suggests the usefulness of this model for the ophthalmic drug delivery of TAC-NMF to the back of the eye via the trans-scleral route [61].

Landi et al. investigated developed a mice model of diabetic retinopathy to analyses the effect of tacrolimus immunosuppressive drug. The researchers treated the mice with 10ug of FK506 per week and analyzed the level of neovascularization and TNF-a, VEGF, iNOS and COX-2 proteins. Tacrolimus treatment was able to significantly lower retinal VEGF-A, TNF-a, iNOS and COX-2, as well as the amount of neovascularization in the streptozotocin-induced diabetic mice [26]. This study demonstrated the effectiveness of tacrolimus treatment in diabetic retinopathy. Kimsa et al. investigated the effect of tacrolimus treatment on RPE cells pretreated with liposaccharide (LPS) for inducing inflammation. Their results showed that the mRNA levels for transforming growth factor b (TGFb) 2 and TGFbR3 were decreased after tacrolimus and LPS treatment. The TGFb family plays a pivotal role in the pathogenesis of various back of the eye diseases [60]. These studies warranted the investigation of TAC-NMF on D407 retinal cells to investigate the inflammatory changes after treating them with inflammatory and ROS-inducing agents such as SI. It was observed that both TAC and TAC-NMF could reduce the levels of ROS in SI-pre-treated D407 cells. To further evaluate the bioavailability and mechanism of inflammation lowering in D407 retinal cells, levels of various cytokines (TNF-α, IL-6 and IL-β) and VEGF-A were analyzed. Here the cells were pretreated with SI retinotoxin. TAC-NMF treatment demonstrated a reduction in TNF-α, IL-6 and IL-β cytokine levels and VEGF-A at 24 and 12 h as compared to the SI treatment group. Moreover, TAC-NMF treatment was also able to lower the ROS in D407 cells pre-treated with SI. The studies can imply that TAC-NMF can not only serve as an excellent carrier for hydrophobic drug, but can also lower the inflammatory cytokines, VEGF-A and ROS in D407 retinal cells.

## 5. Conclusions

In this study, we developed and optimized a clear TAC-NMF solution having potential to be used as an intravitreal injection or a topical formulation for wet AMD. The optimized formulation had a particle size below 20 nm and neutral zeta potential for maximum uptake in ocular cell lines. TAC-NMF also demonstrated excellent trans-well permeability in the in-vitro dual-chamber eye model. Placebo nanomicelles has almost complete cell viability when analyzed on ocular cells. In-vitro biocompatibility assay showed very little generation of inflammatory cytokines in macrophage cell-lines, suggesting the high biocompatibility of TAC-NMF. In-vitro bioassay using retinal cell lines pretreated with retinotoxin like sodium iodide demonstrated a reduction in inflammatory markers and reactive oxygen species after treating with TAC-NMF. These results along with the immunosuppressive effect of tacrolimus can prove to be a beneficial strategy for AMD. This can advocate the application of tacrolimus nanomicellar formulation as a promising strategy for back of the eye disorders.

## Figures and Tables

**Figure 1 pharmaceutics-12-01072-f001:**
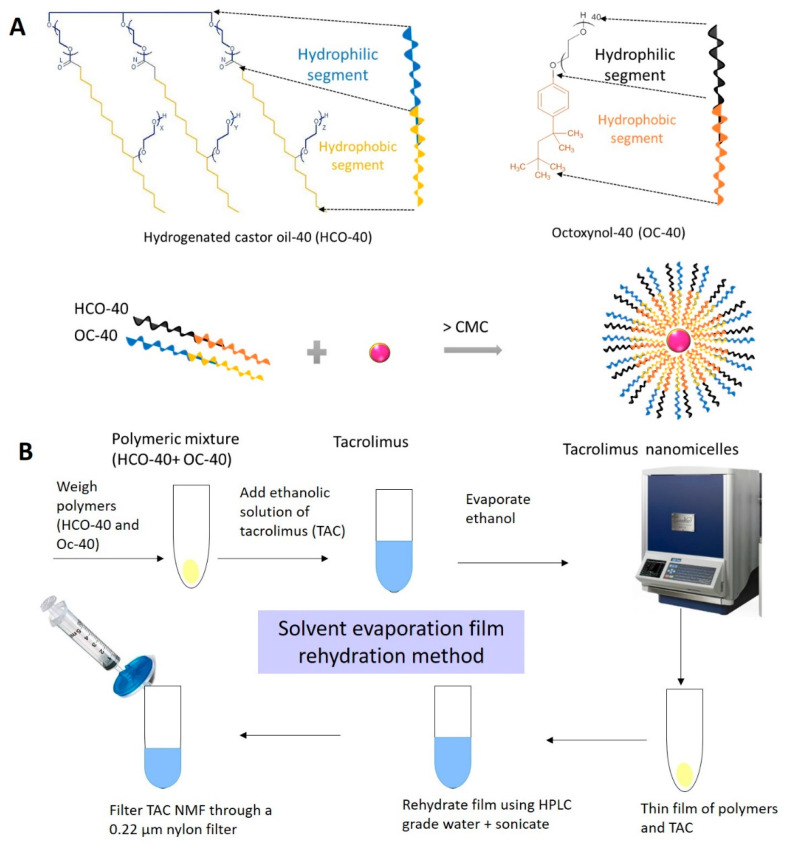
Pictorial representation of tacrolimus nanomicellar (TAC-NMF) formulation development. (**A**) Self-assembly of TAC-NMF nanomicelles from amphiphilic PEG-Hydrodenated castor oil-40 (HCO-40) and Octyxonal-40 (OC-40) amphiphilic polymers above critical micellar concentration (CMC). This allows entrapment of hydrophobic drug tacrilimus in the core of the nanomicelles. (**B**) Schematic representation for prepartaion of TAC-NMF using solvent evaporation-film rehydration method.

**Figure 2 pharmaceutics-12-01072-f002:**
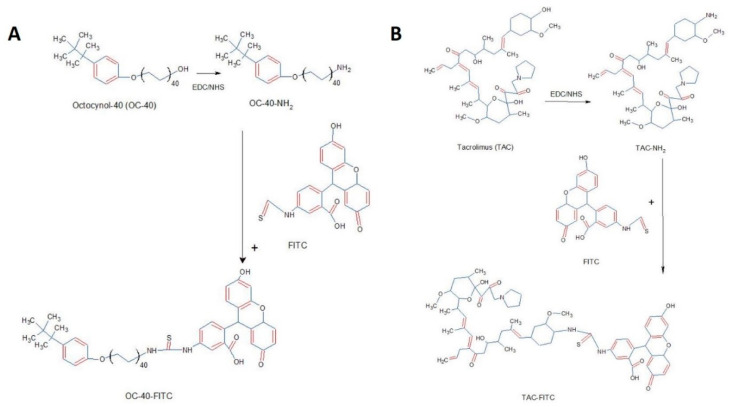
Fluorescein isothiocynate (FITC) conjugation reaction. (**A**) FITC conjugation to OC-40 polymer present in tacrolimus nanomicelles (TAC-NMF) formulation, vis DEC/NHS coupling reaction. (**B**) Tacrolimus (TAC) conjugation reaction to FITC via DEC/NHS coupling reaction.

**Figure 3 pharmaceutics-12-01072-f003:**
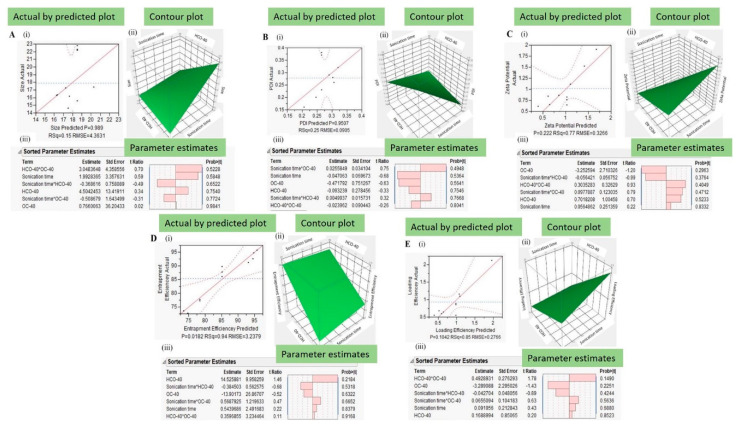
Actual by Predicted plots (i) Contour plots (ii) and Parameter estimates (iii) for dependent variables like Size (**A**), PDI (**B**), Zeta Potential (**C**), Loading Efficiency (**D**) and Entrapment Efficiency (**E**).

**Figure 4 pharmaceutics-12-01072-f004:**
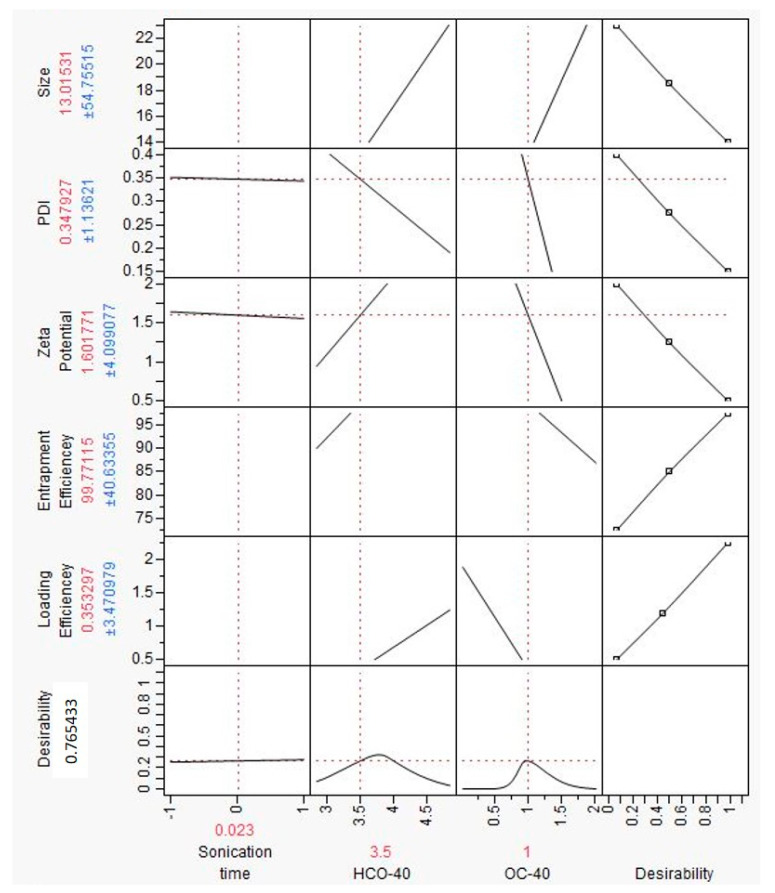
Prediction Profiler for dependent variables such as Size, PDI, Zeta Potential, Loading Efficiency and Entrapment Efficiency depending upon the values of independent variables like polymer concentrations; HCO-40 and OC-40 and sonication time.

**Figure 5 pharmaceutics-12-01072-f005:**
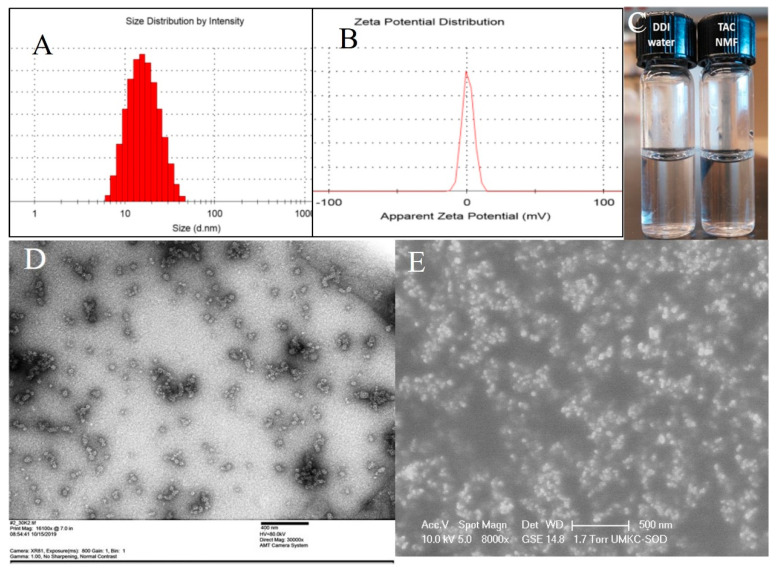
Size distribution of (**A**) Tacrolimus nanomicelles (TAC-NMF) and (**B**) Zeta potential of TAC-NMF, (**C**) Visual comparison of TAC-NMF with DDI water, and (**D**) TEM image of TAC-NMF (**E**) SEM image of TAC-NMF.

**Figure 6 pharmaceutics-12-01072-f006:**
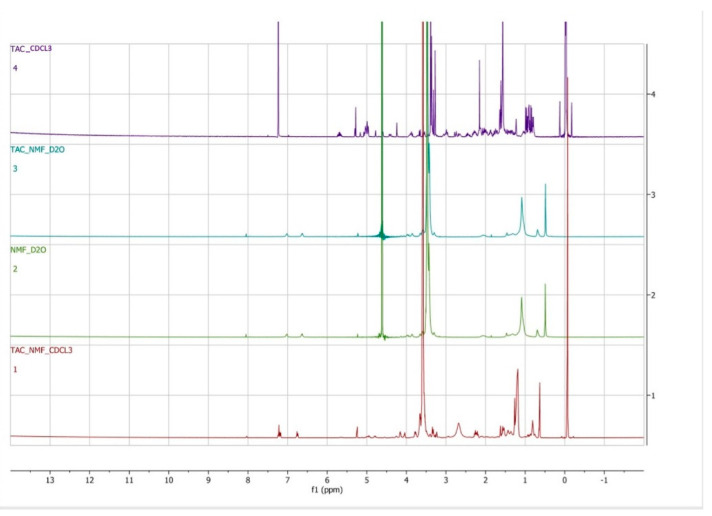
Qualitative ^1^H NMR studies. (1) ^1^H NMR spectrum for TAC drug in CDCl_3_, (2) ^1^H NMR spectrum for TAC-NMF in D_2_O, (3) ^1^H NMR spectrum for blank NMF in D_2_O and (4) ^1^H NMR spectrum for TAC-NMF in CDCl_3._

**Figure 7 pharmaceutics-12-01072-f007:**
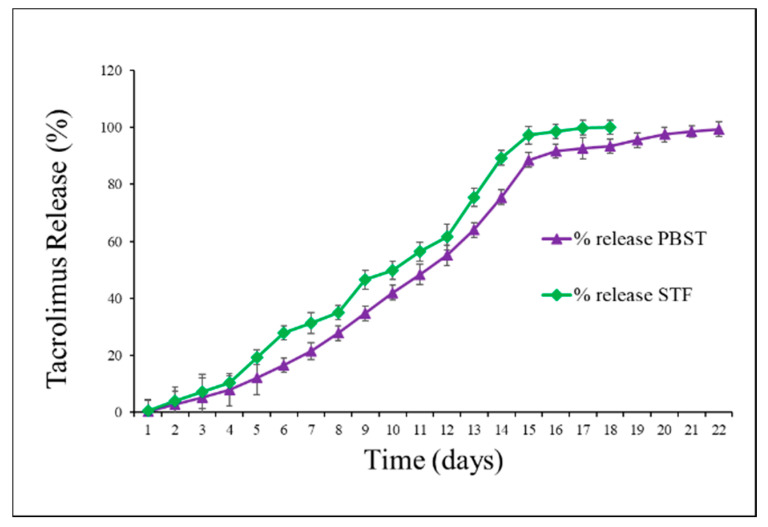
In-vitro release of Tacrolimus (TAC) from Tacrolimus nanomicellar formulation (TAC-NMF) evaluated in PBST and STF buffer solutions.

**Figure 8 pharmaceutics-12-01072-f008:**
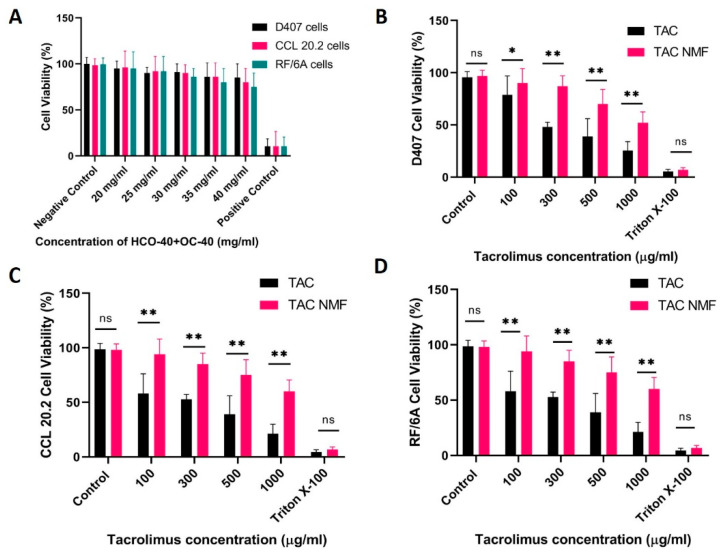
Cell viability determination by MTT assay. (**A**) Cell viability (%) of D407, CCL 20.2 and RF/6A cells after treatment with placebo NMF composed of mixture of HCO-40 and OC-40 amphiphilic polymers ranging from 20–40 mg/mL of total HCO-40 and OC-40 in a fixed ratio of (HCO-40:OC-40: 3.5:1) for 24 h. (**B**) Cell Viability (%) of retinal pigment epithelium cells (D407) after treatment with TAC and TAC-NMF for 24 h. (**C**) Cell Viability (%) of conjunctival cells (CCL 20.2) after treatment with TAC and TAC-NMF for 24 h. (**D**) Cell Viability (%) of choroidal endothelial cells (RF/6A) after treatment with TAC and TAC-NMF for 24 h. (* *p* ≤ 0.05 and ** *p* ≤ 0.01 as compared to the corresponding control group, here TAC).

**Figure 9 pharmaceutics-12-01072-f009:**
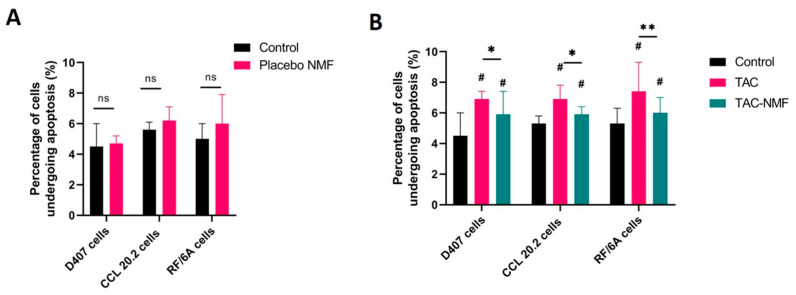
Determination of cell apoptosis in D407, CCL 20.2 and RF/6A ocular cells after treatment with placebo NMF, TAC and TAC-NMF by Annexin V/FITC and PI-Staining. (**A**) Percentage of cell apoptosis in D407, CCL 20.2 and RF/6A cells after treatment with placebo NMF for 24 h determined by fluorescence assisted cell sorting (FACS). (**B**) Percentage of cell apoptosis in D407, CCL 20.2 and RF/6A ocular cells treated with TAC or TAC-NMF for 24 h determined by FACS. Results are expressed as a mean of three independent experiments, *n* =3 ± SD. (# *p* ≤ 0.05 as compared to control, * *p* ≤ 0.05 between group and ** *p* ≤ 0.01 between group, ns = not significant difference between groups).

**Figure 10 pharmaceutics-12-01072-f010:**
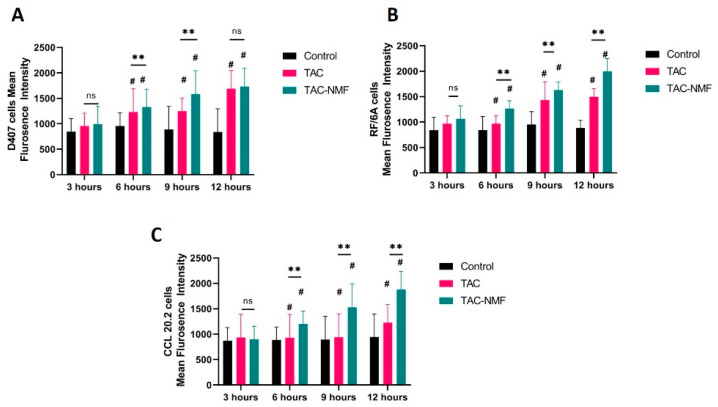
Time dependent uptake and quantification of FITC-labelled TAC solution and FITC-labelled TAC-NMF solution in (**A**) D407, (**B**) RF/6A and (**C**) CCL 20.2 cells using fluorescence assisted cell sorting (FACS). Results are expressed as a mean of three independent experiments, *n* = 3 ± SD. (# ≤ 0.05 as compared to control and ** *p* ≤ 0.01 between group, ns = not significant difference between groups).

**Figure 11 pharmaceutics-12-01072-f011:**
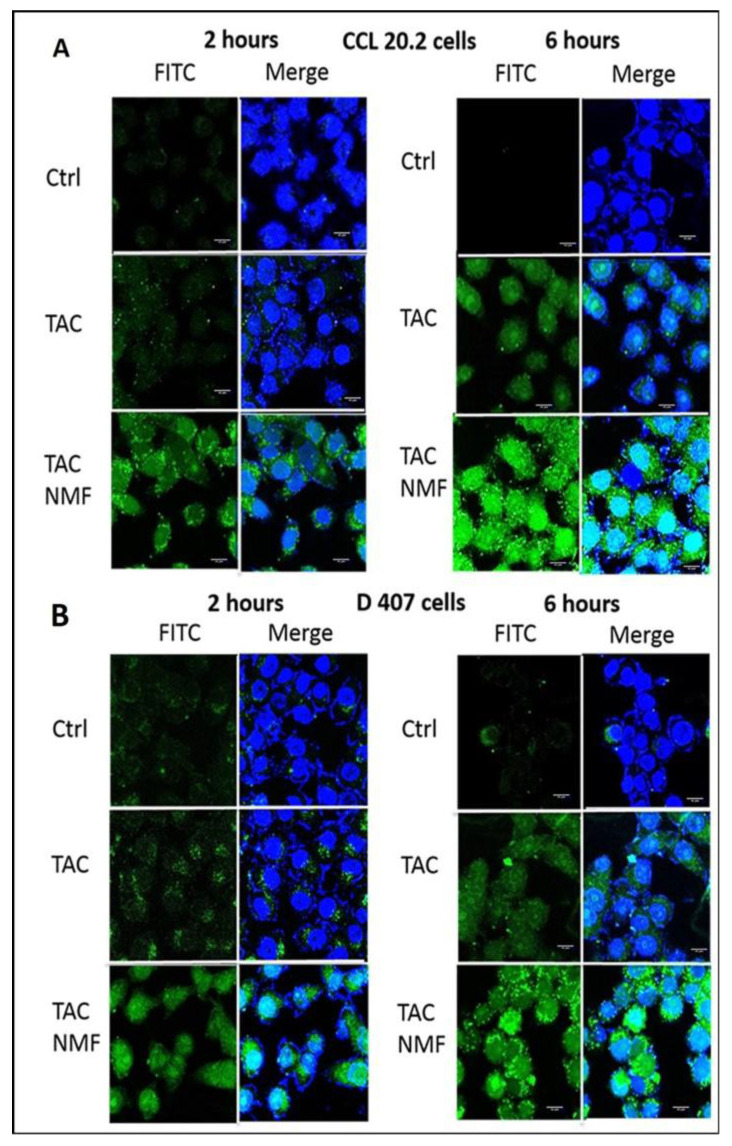
Time dependent uptake of FITC-labelled TAC solution and FITC-labelled TAC-NMF solution in (**A**) CCL 20.2 and (**B**) D407 cells CLSM.

**Figure 12 pharmaceutics-12-01072-f012:**
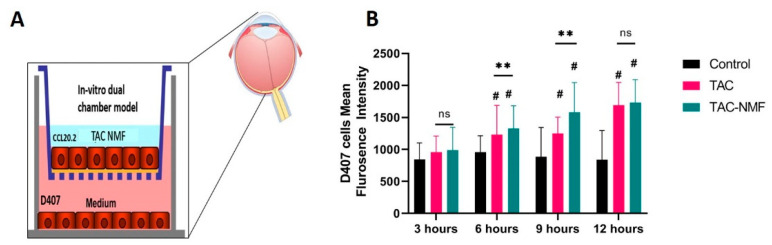
In-Vitro Cellular Transport of TAC-NMF. Schematic representation of the preparation of (**A**) in-vitro conjunctival-retinal dual chamber model and FACS analysis of time dependent in-vitro transport of FITC-labelled TAC-loaded nanomicelles in Transwell diffusion chamber with (**B**) CCL 20.2 and D407 cell lines. Results are expressed as a mean of three independent experiments, *n* = 3 ± SD. (# ≤ 0.05 as compared to control and ** *p* ≤ 0.01 between group, ns = not significant difference between groups).

**Figure 13 pharmaceutics-12-01072-f013:**
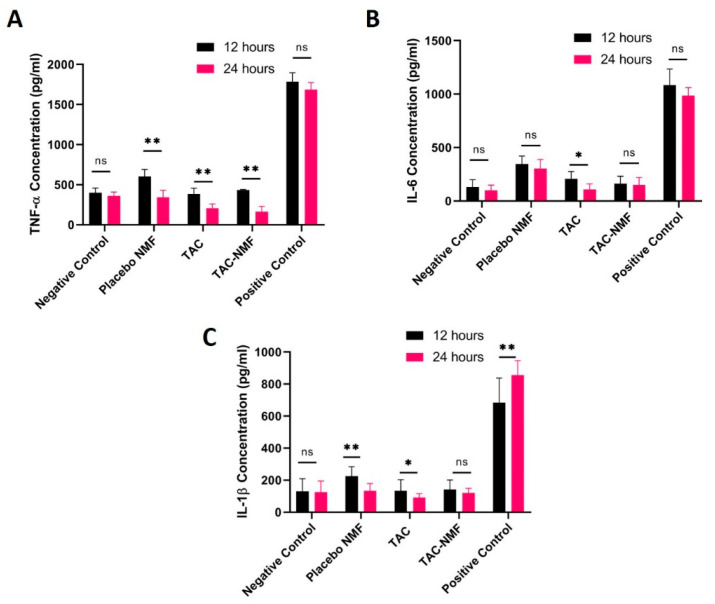
In-vitro release of TNF-α (**A**), IL-6 (**B**) and IL-1β (**C**) and from RAW 264.7 cells following 12 and 24 h exposure of tacrolimus (TAC), placebo NMF and TAC-NMF using ELISA. Results are expressed as mean of three independent experiments, *n* = 3 ± SD. (* *p* ≤ 0.05 as compared to 12 h ** *p* ≤ 0.01 as compared to 12 h, ns = not significant difference between groups).

**Figure 14 pharmaceutics-12-01072-f014:**
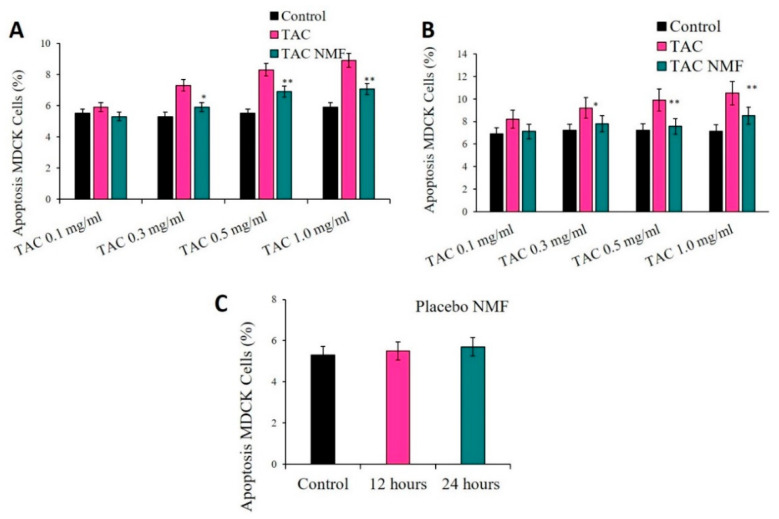
In-vitro biocompatibility study of TAC-NMF in MDCK kidney cells by determining by Annexin V/FITC and PI-Staining. MDCK cells were treated for 12 (**A**) and 24 (**B**) hours with TAC and TAC-NMF at concentrations of tacrolimus at 0.1, 0.3, 0.5 and 1.0 mg/mL. MDCK cells were also treated with placebo NMF (**C**) for 12 and 24 h and analyzed for apoptosis by Annexin V/FITC and PI-Staining. Results are expressed as mean of three independent experiments, *n* = 3 ± SD. (* *p* ≤ 0.05 as compared to TAC ** *p* ≤ 0.01 as compared to TAC).

**Figure 15 pharmaceutics-12-01072-f015:**
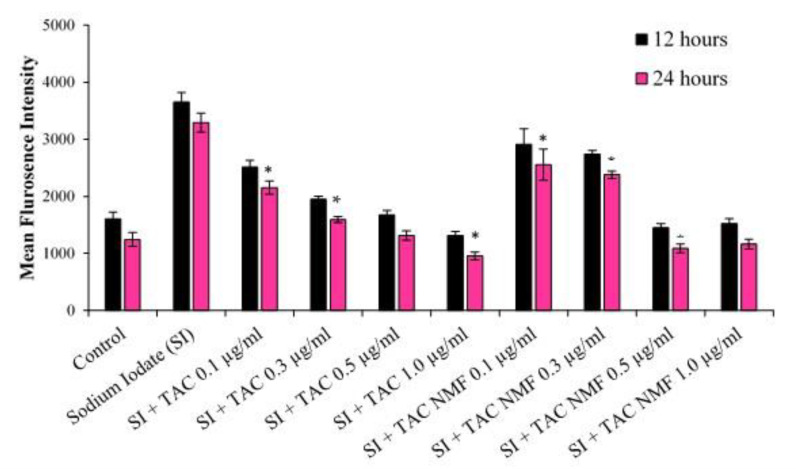
Evaluation of formation of Reactive Oxygen Species (ROS) in D407 retinal pigment epithelium cells pretreated with Sodium Iodate (SI) after treatment with TAC-NMF and TAC by DCFDA Assay. D407 cells treated with SI 10 ug/mL for 6 h followed by treatment of TAC (0.1, 0.3, 0.5, 1.0 ug/mL) and TAC-NMF (0.1, 0.3, 0.5, 1.0 ug/mL) for 12 and 24 h. After the treatment, ROS was measured by DCFDA dye and the ROS was determined by FACS. Results are expressed as mean of three independent experiments, *n* = 3 ± SD. (* *p* ≤ 0.05 as compared to 12 h * *p* ≤ 0.05 as compared between groups).

**Figure 16 pharmaceutics-12-01072-f016:**
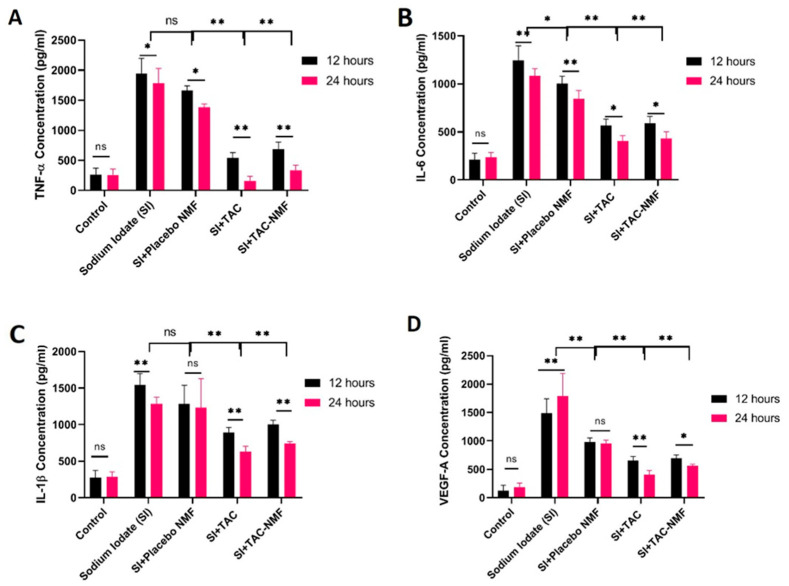
In-vitro release of TNF-α (**A**), IL-6 (**B**), IL-1β (**C**) and VEGF-A (**D**) from D407 cells after treatment with Sodium Iodate (SI) for 6 h following 12 and 24 h exposure of TAC drug, placebo NMF and TAC-NMF using ELISA. (* *p* ≤ 0.05, ** *p* ≤ 0.01, ns = not significant as compared to the corresponding control group, here 12 h TNF-α release, 12 h IL-6 12 h IL-1β and 12 h VEGF-A release).

**Table 1 pharmaceutics-12-01072-t001:** Full factorial design of experiment by JMP software. Analysis of independent variables like sonication time, HCO-40 wt % and OC-40 wt % on dependent variables like size, polydispersity index (PDI), zeta potential, entrapment efficiency (EE) and loading efficiency (LE).

Formulation	Coded Design			Uncoded Design		
	X1	X2	X3	X1 = Sonication Time (Minutes)	X2 = HCO-40 (wt %)	X3 = OC-40 (wt %)
F1	+	+	+	25	3.5	3
F2	−	+	+	20	3.5	3
F3	0	0	0	20	2	2
F4	+	−	−	25	0.5	1
F5	+	−	+	25	0.5	3
F6	−	+	−	20	3.5	1
F7	0	0	0	22.5	2	2
F8	−	−	+	20	0.5	3
F9	−	−	−	20	0.5	1
F10	+	+	−	25	3.5	1
F11	0	0	0	22.5	2	2

X1-sonication time (minutes), X2-HCO-40 (wt %) and X3-OC-40 (wt %).

**Table 2 pharmaceutics-12-01072-t002:** Full factorial design of experiment by JMP software. Analysis of independent variables like sonication time, HCO-40 wt % and OC-40 wt % on dependent variables like size, PDI, zeta potential, entrapment efficiency (EE) and loading efficiency (LE).

Formulation	Pattern	Sonication (min)	HCO-40 wt %	OC-40 wt %	Size (nm)	PDI	Zeta (mV)	%EE	%LE
F1	+++	25	3.5	3	17.22	0.29	0.658	94.74	0.59
F2	−++	20	3.5	3	14.59	0.2	0.837	92.49	0.55
F3	000	20	2	2	22.15	0.37	0.716	86.05	0.85
F4	+−−	25	0.5	1	16.3	0.37	1.292	77.22	2.25
F5	+−+	25	0.5	3	16.17	0.28	1.518	77.67	1.15
F6	−+−	20	3.5	1	15.41	0.25	0.512	97.13	0.72
F7	000	22.5	2	2	22.19	0.38	0.684	89.72	0.89
F8	−−+	20	0.5	3	16.35	0.32	1.081	73.53	10.9
F9	−−−	20	0.5	1	13.34	0.26	1.896	72.71	2.12
F10	++−	25	3.5	1	15.57	0.16	0.611	91.17	0.68
F11	000	22.5	2	2	22.72	0.23	0.819	95.58	0.95

HCO-40: hydrogenated castor oil-40, OC-40: Octyxonyl-40, PDI: poly dispersity index, EE: entrapment efficiency, LE: loading efficiency.

**Table 3 pharmaceutics-12-01072-t003:** Tacrolimus nanomicellar formulation (TAC-NMF) dilution study.

Dilution Factor	Hydrodynamic Size (nm)	Polydispersity Index (PDI)	Zeta Potential (mV)
0	15.36	0.183	0.522
10	15.78	0.251	0.651
20	16.09	0.270	0.72
40	16.32	0.313	0.86
50	16.46	0.259	1.016
100	16.72	0.322	1.13
150	16.85	0.468	1.523
200	17.01	0.672	1.92

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
