# Peer review of "Self-Assembling Tacrolimus Nanomicelles for Retinal Drug Delivery"

_pharmaceutics, 2020, doi:10.3390/pharmaceutics12111072_

Round 1

Reviewer 1 Report

This is an extensive in vitro study about the effect of Tacrolimus (TAC) encapsulated nanomicelles  on different ocular cell lines. The authors used a systematic approach to produce optimal nanomicelles (NMF) from a combination of the 2 polymers HCO-40 and OC-40.  TAC and TAC-NMF were then tested for cytotoxicity in different ocular cell lines (RPE line D407, retinal choroidal endothelial cell line RF/6A and human corneal epithelium CCL20.2). Encapsulated TAC (TAC-NMF) was less toxic and more efficiently taken up than TAC alone. It was also transported over a barrier as shown in a transwell assay. Biocompatibility (toxicity and production of cytokines) was also tested in RAW 246.7 macrophage cells and in MDCK kidney cells. The data show that TAC-NMF and TAC alone block release of cytokines, and that TAC-NMF is less toxic than TAC alone. However, placebo NMF (nanomicelles alone) also kept the apoptosis rat at the same level as the untreated control, and reduced cytokine levels, although not to the same extent.  D407 (RPE) cells were treated with sodium iodate (SI) to induce formation of reactive oxygen species and then ttreated with TAC or TAC-NMF of different concentrations. .Higher concentrations of TAC or TAC-NMF brought oxidative stress down to control levels which was also shown by a cytokine panel..

General comments:

This is a well-designed study with many important data. However, the presentation needs to be improved. The text requires some reorganization. Part of the discussion is mixed in with the results. On the other hand, the discussion just sums up the results without discussing the literature.

Frequently, the authors state “ocular cells like D407, CCL 20.2 and RF/6A”. It would be better to change this to “ocular cell lines D407, CCL 20.2 and RF/6A”  - omit “like”. Please change this throughout the text.

Specific comments:

Introduction:

p.2, line 95: “The current treatment for AMD…” – please add “wet” or “exudative” AMD. There is no treatment for dry AMD.

Materials and Methods:

  1. 5, line 202: “HPLC method for determining …” – add “was used”. Otherwise this is not a complete sentence.

p.5, line 216: “pelette” – should be “pellet”

p.6, line 248: Figure 3 is cited before Figure 1.

p.6, line 264: “0.22 µm nylon” – add: “filter”

p.7, line 296: add “and” between “tubes” and “centrifuged”.

p.7, line 305: take out comma between “time” and “point”.

p.7, line 305: add comma after “(3x5 minutes)” and continue in lower case.

p.8, line 349: “The cells suspension was acquired by FACS” -  proposal: “The cells’ fluorescence was analyzed by FACS”

p.8, line 255: typo “ELIZA” – should be “ELISA”

Results:

p.9, line 388: “compares the variation occurs” – better: “compared the variation that occurs”

p.13, line 506: “… could form stable …” – should be: “could be more stable”

p.14, line 528: add comma after “For this purpose”

Discussion:

Too short, part of the text from results should go into discussion (see general comments)

Table 2 (p.13): maybe should highlight F6, the formula that was ultimately used with bolding or other highlights.

Figures:

Figure 1: This figure is difficult to follow. It is difficult to read the small details. Although it is written in the figure legends, it would be helpful to add headings to the different panels to instantly see what is being looked at.

Figure 11: it would be helpful for the Reader if placebo NMF, TAC, and TAC-NMF could be separated visually by using different colors. Otherwise the different panels are difficult to distinguish. The same applies to Figures 13 and 14 (distinguish Tac and TAC-NMF by color).  

Reviewer 2 Report

Comments: Authors have succeed to develop a topical eye drop treatment using nanomicellar technology. It is interesting work can be delivered into market later soon.  However, there are still unclear points required a more solid study design.

  • Authors described the structure of their micelles composed of mixture of amphiphilic polymers like Hydrogenated castor oil-40 (HCO-40) and Octyxonyl-40 (OC-40). I am worried that  Hydrogenated castor oil-40 is used in form of PEG-Hydrogenated castor oil-40 and  Octyxonyl-40 is used as surfactant -- solubilizing agents. Authors should to clarify the structure.
  • The zeta potential results showed the net charge of micelles in all forms are slightly positive However, mV value for all forms  is very weak. This could not produce  repulsive force among individual particles  leading to increase their aggregation. Authors should have to care  more of their design, instead of  obtaining many formulations.
  • Authors didn’t explain scientifically why F6 was chosen as optimized formulation although its zeta potential is slightly positive with less value than other formulations.
  • Authors should have introduced schematic structure illustrated the possible interaction of entrapment of Tacrolimus
  • Authors should  have used   formal shape  over all the  TEXT  (for examples, in line 289, REF (33,34), in line 424 F6  or you can use  F-6 in line 428)
  • Many English typing error should be corrected over all the text.
  • Section 2.8.1 FITC labelling . Author explained labelling micelles and free drug with FITC but the reaction is still unclear because FITC includes isothiocyanate group that can be reacted in the presence of amino group of polymer. In the formula structure of micelles, there is no any amino group attached . However,  there is one case could be done that FITC is attached in presence of  EDAC/NHS or maleimide.  Authors should have explained the chemical interaction of such this conjugation.  
  • In section 3.7.1 MTT assay. Authors revealed that MTT assay was investigated after 24h. This is also unclear condition. MTT assay should be investigated after 36h because the micelles have oily structure .This oil is a mixture of saturated and unsaturated fatty acid esters linked to a glycerol. This fatty acid takes time to be digested by lipase enzymes inside lysosomes .  
  • The authors should follow the format model of reference in journal style as follow (Author 1, A.B.; Author 2, C.D. Title of the article. Abbreviated Journal Name Year, Volume, page range.
  • More than 26 references are down 2015- These reference should be updated
  • It is recognized that the manuscript results were published “ https://www.eventscribe.com/2019/PharmSci360/fsPopup.asp?efp=SUlFUEhHSFQ4MDkx&PosterID=234587&rnd=0.491479&mode=posterinfo. Authors should have cited this in their text

Reviewer 3 Report

In this paper author has reported nano micellar formulation of tacrolimus using combination of two polymers; hydrogenated castor oil and octyxonyl-40, formulation to be used as intravitreal injection for treatment of age-related macular degenerative (AMD) by decreasing ROS and pro-inflammatory cytokine in retinal cells. In conclusion, author found that the formulation was clear, stable and effective at lower concentration of polymer mixture. It can be delivering its therapeutic effect against AMD. This research can be suitable for publication subjected to major revision.

  1. In introduction “Although highly………………non patient compliant” looks incomplete and need to be recheck.
  2. Figure 5. “In-vitro release of Tacrolimus (TAC) from Tacrolimus nano micellar formulation (TAC-NMF) evaluated in PBST and STF buffer solutions” need to redraw to remove -20.
  3. In discussion part appropriate references need to be cited where required.
  4. Section 3.12 need to revise.
  5. Following studies need to be added to better justify the data:

(i) SEM

(ii) Rheological studies

(iii) Chemical stability study of tacrolimus-loaded polymeric micelles

(iv) Compatibility study between drug and polymer

Round 2

Reviewer 1 Report

The authors have considerably improved the manuscript, especially the figures. They have added 2 explanatory figures (Figure 1 and 2) and improved the remaining figures by adding colors and more labels, and simplifying some (reducing amounts of panels) so they are now more readable.

They added more descriptions and a figure of the FITC labeling (section 2.8.1).

The discussion has been expanded as requested.

The reviewer has only a few comments.

Figure legend of Figure 1 (p. 6, line 202): typo – “represnentation”

  1. 8, line 268: “palette” – should be “pellet”?
  2. 9, line 331: “were seeded” is repeated twice in the sentence.
  3. 20, line 659: “that the controlled release of the drug can be responsible for making tacrolimus less potent …” – proposal: “… less toxic …”

Author Response

The authors highly appreciate the comments of the reviewers about the revised manuscript, specifically about improved figures and better explanations. 

Figure legend of Figure 1 (p. 6, line 202): typo – “represnentation”

This is now corrected. 

  1. 8, line 268: “palette” – should be “pellet” This is now corrected. 
  2. 9, line 331: “were seeded” is repeated twice in the sentence. This is now corrected. 
  3. 20, line 659: “that the controlled release of the drug can be responsible for making tacrolimus less potent …” – proposal: “… less toxic …” "Less toxic " instead of "less potent" is now included in the text. 

Reviewer 2 Report

The comments were mostly answered and manuscript is more clear now. However, It is still references are not designed in journal format like this "Author 1, A.B.; Author 2, C.D. Title of the article. Abbreviated Journal Name Year, Volume, page range. 

Author Response

The comments were mostly answered and manuscript is more clear now. However, It is still references are not designed in journal format like this "Author 1, A.B.; Author 2, C.D. Title of the article. Abbreviated Journal Name YearVolume, page range. 

All the references are now in the above-mentioned format. 

The authors highly appreciate the reviewers comments and suggestions.

Reviewer 3 Report

Everything looks fine in the revised manuscript. However, as informed by author that SEM analysis is a difficult task. However, I believe that it is not a difficult task. Multiple reports confirm that lyophilization of nanomicelles does not create such a problem. Thus, it will be better to add SEM data before final acceptance.

Author Response

Everything looks fine in the revised manuscript. However, as informed by author that SEM analysis is a difficult task. However, I believe that it is not a difficult task. Multiple reports confirm that lyophilization of nanomicelles does not create such a problem. Thus, it will be better to add SEM data before final acceptance.

SEM Analysis of TAC-NMF is now added in the revised manuscript. Interestingly the image displays spherical particles less than 50 nm. This was a valuable suggestion which will definitely improve the quality of the manuscript. 

Thank You.